# Expectation generation and its effect on subsequent pain and visual perception

Rotem Botvinik-Nezer[1,2*], Stephan Geuter[3], Martin A. Lindquist[3], Tor D. Wager[2*]

**1** Department of Psychology, The Hebrew University of Jerusalem, Jerusalem, Israel, **2** Department of Psychological and Brain Sciences, Dartmouth College, Hanover, New Hampshire, United States of America, **3** Department of Biostatistics, Johns Hopkins Bloomberg School of Public Health, Baltimore, Maryland, United States of America

\* tor.D.Wager@dartmouth.edu (TDW); rotem.botvinik-nezer@mail.huji.ac.il (RB-N)

## Abstract

Bayesian accounts of perception, such as predictive processing, suggest that perceptions integrate expectations and sensory experience, and thus assimilate to expected values. Furthermore, more precise expectations should have stronger influences on perception. We tested these hypotheses using a within-subject paradigm with social cues consisting of what participants were told were ratings from 10 prior participants, but which were actually constructed to independently manipulate the cue mean, variance (precision), and skewness independent of the actual stimulus intensity delivered. Forty-five participants reported their expectations regarding the painfulness of thermal stimuli or the visual contrast of flickering checkerboards. In a second session, similar (sham) cues were each followed by either a noxious thermal or a visual stimulus. Perceptions assimilated to cue-based expectations in both modalities, but precision effects were modality-specific: more precise cues enhanced assimilation in visual perception only, while higher uncertainty slightly increased reported pain. fMRI analysis revealed that the cues affected higher-level affective and cognitive systems--including assimilation to the cue mean in the Stimulus Intensity-Independent Pain Signature-1 (SIIPS-1), a neuromarker of endogenous pain processing--and in the nucleus accumbens. There were no predictive cue effects on the Neurological Pain Signature (NPS), a neuromarker of nociceptive pain. Region of interest analyses showed activity consistent with aversive prediction-error-like encoding in the periaqueductal gray during pain perception, but no cue or prediction error-related responses in early perceptual processing systems. Furthermore, behavioral and computational models of the expectation session revealed that expectations construction was biased towards extreme cue values in both modalities, and towards low-pain cues specifically. These findings suggest that predictive processing theories should be extended with mechanisms such as selective attention to outliers, and that expectation generation and its perceptual effects vary by sensory modality and primarily influence higher-level processes rather than early perception, at least when cues are not reinforced.

**Data availability statement:** Processed data and analysis codes are publicly shared on Github: https://github.com/rotemb9/PPRI-paper (release v2.0.0). CANlab neuroimaging analysis tools that were used as part of the analysis are available at https://canlab.github.io/. The imaging data are publicly shared on OpenNeuro: doi: https://doi.org/10.18112/openneuro.ds005413. v1.0.0.

**Funding:** This study was funded by the National Institutes of Health (NIH; R37MH076136 TDW, R01MH129397, and R01EB016061 to MAL and TDW). The funders had no role in study design, data collection and analysis, decision to publish, or preparation of the manuscript.

**Competing interests:** The authors report no competing interests.

## Author summary

Expectations shape how we perceive the world, influencing both what we feel and what we see. But how do we form these expectations, and how do they affect our perception? We investigated how people generate expectations about upcoming pain and visual stimuli based on complex multi-attribute social cues—ostensible ratings of 10 other participants—and how these expectations influence perception and brain activity. Using behavioral experiments and computational modeling, we found that people rely heavily on an average summary of complex cue features when forming expectations, but extreme values also play a role, especially in lowering pain expectations when extreme values signal lower pain. Perceptual judgments in both modalities tended to shift toward expected values, but the uncertainty of these expectations affected perception in different ways across pain and vision, and were only consistent with precision-weighted integration of cue information posited by predictive processing accounts in vision. Our neural findings suggest that expectations do not directly alter early sensory processing but instead shape how information is interpreted at later decision-making stages. These insights enhance our understanding of how the brain integrates contextual cues and may inform better approaches to pain management and sensory disorders.

## Introduction

Current theories of perception, such as Bayesian theories [1,2] and predictive processing [3,4], posit that the brain represents the world with an internal generative model, used to predict future external (sensory input) and internal (physiological states) events. Model and cue-based expectations (prior beliefs) computationally combine with incoming sensory information (likelihoods) to form perceptions (posterior beliefs) [5,6]. In line with this principle, predictions about upcoming stimuli have been shown to shape their perception across sensory modalities. For example, reported painfulness of noxious stimuli assimilates to expected values [7–12]. Likewise, the perceived direction of randomly moving dots is biased towards expected directions [5,13,14]. Such assimilation is beneficial, as it improves perception when contextual information is relevant. Similar Bayesian principles have been suggested to underlie interoception [15–17] and diverse aspects of cognition [18,19] including planning and decision-making (e.g., active inference [20,21]). However, such models also have contrastive mechanisms, which drive learning. While assimilation of perceptions to predicted values is one way to resolve discrepancy between predictions and sensory input, another important way is to update the predictive model to match the input, i.e., to learn. Prediction error (PE)-driven updating is a crucial form of learning. However, which brain signals assimilate to expectations, perhaps forming a neural substrate for perception, and which encode PEs and thus drive learning, is a topic of active investigation [5,22–30].

Bayesian models of predictive processing make predictions about how the relative precision of cues and sensory stimuli affects perception and neural responses. Prior predictions and likelihoods (i.e., the expectation and the incoming sensory information) are weighted by their relative precision ("precision weighting") [2,5,6,30–32]. Thus, Bayesian models predict an interaction between the expected value and its precision, such that more certain expectations (prior beliefs) should affect perception (posterior beliefs) more strongly. This hypothesis has been supported by studies on pain perception showing stronger placebo analgesia in individuals with more precise treatment expectations [25,33], and greater assimilation of perceived pain towards expectations with higher cue precision [34,35]. Conversely, other studies have found that lower precision or predictability (higher uncertainty) leads to more pain, putatively because uncertainty is aversive in the context of potential harmful stimuli [7,36]. Some other studies did not replicate these effects and even found instead that more certain cues led to more pain [37–41]. Finally, a recent meta-analysis of 70 studies did not find an effect of predictability (including uncertainty about timing, duration, or location) on pain perception [41]. Instead, meta-regression analyses showed that pain was higher for whichever condition (unpredictable vs. predictable) elicited higher pain expectancy and state anxiety, implying that expectations may confound predictability manipulations in some studies. The contradicting findings regarding the effect of the cue certainty on pain perception [42] may stem from several factors, such as experimental design and sensory modality. For examples and descriptions of how we address them in the current study, see Table 1.

A promising paradigm, which we adopt here, uses cues with rich information about the distribution of predicted values, allowing within-person manipulation of precision and other distributional properties [7,8,10,34,37]. This can substantially increase power and reduce confounds in testing manipulations of precision. The version that we use presents multiple-valued cues as ratings of multiple previous participants (Fig 1), providing an additional advantage in that it provides a plausible type of social observational learning that may be particularly impactful [43–48]. Though such cues have been used successfully in multiple studies, an additional complexity is that manipulation of the cues' precision–i.e., the variance of the presented cue values–influences other properties, including the maximum and minimum values. If people weigh all cue values equally, this is ignorable, but participants may instead attend to the highest or lowest values, depending on the task and their predispositions (e.g., if some participants find pain to be threatening and are vigilant for high-pain cues).

Such effects are not taken into consideration in most studies testing Bayesian accounts of perception, but there is evidence that they can be important. For instance, a previous study in visual perception has demonstrated overweighting of inliers (values closer to the mean) [49]. Such effects may also be sensitive to the precise demands of the task. For example, when asked to determine which series of numbers was on average larger (via visual or auditory perception), people have been found to overweight outliers (extreme values), specifically larger values [50]. Furthermore, this kind of nonlinear

**Table 1. Factors that could explain previous contradicting results.**

| Factor | Examples | How previous studies differ | How the current study addresses it |
|---|---|---|---|
| Experimental design | Precision manipulation | Previous studies may have unintentionally manipulated additional statistical properties of cues (e.g., outliers, skewness) when altering precision. | This study systematically tests the independent effects of cue mean, variance, and skewness to isolate their contributions. |
| Sensory modality | Pain vs. other stimuli | Uncertainty may be more aversive in pain studies due to its threat value, leading to different effects across sensory modalities. Most studies focus only on one sensory domain. | This study employs a within-subject design to compare uncertainty effects in both pain and visual perception. |
| Participants | Individual differences, trust in cues | Variability in how participants perceive uncertainty (e.g., aversiveness), their self-perception (e.g., pain sensitivity), and trust in provided information can lead to mixed results. | Our sample is based on a specific population like previous studies, and we do not directly manipulate individual differences, but some are measured (e.g., state anxiety, fear of pain, pain catastrophizing). Moreover, the study accounts for trust in the cues by tracking cue effects over time to ensure their believability. |

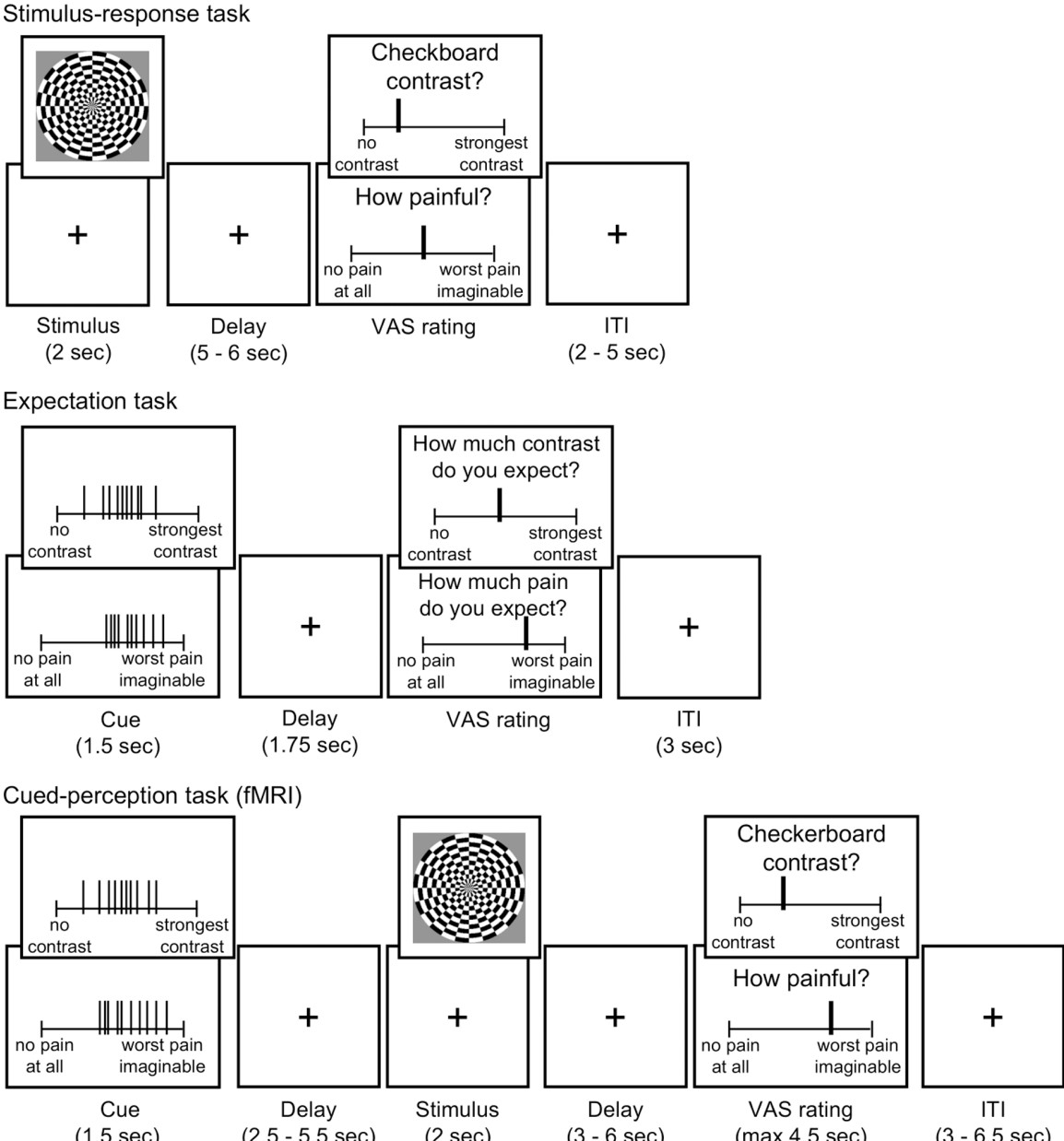

**Fig 1. Experimental design.** An illustration of the three experimental tasks: stimulus-response task (top panel), expectation task (middle panel), and cued-perception task (bottom panel; performed during an fMRI scan). When two rectangles are presented on top of each other (e.g., for the ratings in all tasks), the lower one illustrates pain trials and the upper one illustrates visual trials.

weighting might depend on the sensory modality, and has not been studied in the context of pain perception. If attention to extreme values is related to vigilance for potential threat, people might weight them more strongly for pain than for other, safer modalities. This could, in principle, explain previous findings of higher pain ratings following less precise cues [7]: If people are biased towards maximal pain values ("danger signals"), and higher variance entails larger maximal values, then this effect is expected. Moreover, pain perception is sometimes considered more subjective and ambiguous compared to other types of perception, like perception of visual contrast. Thus, its likelihood might be less precise, and

predictive cue effects might be increased compared to other modalities. If this is the case, it suggests that the nature of the predictive processes at play are at least to some degree modality-specific, and theories of perception must identify differences and commonalities across different types of sensory input [31].

Finally, it is still unclear how expectations, and their precision, influence brain processes underlying pain and visual perception. Some evidence suggests that influences of expectations can reach the earliest stages of sensory processing, for example in the spinal cord [51–53], cortical nociceptive pain processing regions [9,54], and primary visual cortex [14,55]. Other studies show that expectations modulate higher-level affective and cognitive processes [56,57]. However, others have questioned how strongly context-based influences affect perception [58]. Moreover, we have recently shown that placebo treatment (which is thought to operate via expectation modulation, among other effects [59–61]) induces analgesia via modulation of affective and cognitive processes, rather than nociceptive pain processing [62]. If expectation effects mostly operate via high-level affective and cognitive processes, they should be reduced for basic perception that is not subject to affective evaluation, like brightness or visual contrast. Alternatively, these processes could be mediated by different mechanisms across modalities, and thus subject to different behavioral influences and related to different patterns of neural activity.

Here, we test how expectations are generated from distribution cues with varying levels of mean, variance, and skewness (and thus the presence and direction of outliers), and how they affect subsequent perception of visual and painful stimuli (for the study design and illustration of the cues see Fig 1 and Methods). The cues were not externally reinforced (i.e., not predictive of subsequent stimulus intensity), and thus were informational but not subject to associative learning (i.e., they were not classically conditioned). They also did not depend on the participant's individually calibrated intensities, similar to several previous studies [8,10,63,64] (we note that in some other studies, cues were based on the individual's psychometric curve [7,9,37]). Such cues allow us to focus on effects of cognitive predictive information, rather than classical conditioning, which can occur when cues are associated with different levels of subsequent stimulus intensity and are thus truly informative about future intensity. Furthermore, although expectations are influenced by various factors, including cues, prior experience [40,65], and fluctuations in attention [66] among other factors, here we focus specifically on cue-based expectations. By comparing pain and visual perception, we identify common and distinct mechanisms for expectation formation and modulation of perception for stimuli that are potentially harmful (painful) and those that are non-harmful and affectively neutral (visual contrast). The visual modality was chosen because predictive coding theories originated in this domain [67] and have been extensively studied within it [5]. Additionally, as noted earlier, nonlinear weighting has been demonstrated in visual perception. Using fMRI and a priori neuromarkers and individual regions of interest (ROIs), we test whether and how neural responses during pain and visual perception are affected by the different properties of the cue and by the cue-based expectations.

## Results

In the first task, participants rated the painfulness of thermal stimuli and the visual contrast of flickering checkerboards with five levels of intensities per modality (45°C, 46°C, 47°C, 48°C and 49°C, and 5%, 27.5%, 50%, 72.5%, and 95% luminance contrast; for more details see Methods and Fig 1), and 50 trials overall (5 trials per intensity level of each contrast). We tested the effect of the stimulus intensity on the rating with a separate mixed-effects model per modality, controlling for the trial number. The trial number covariate modeled variation in outcomes (pain or visual intensity) over time. As expected, participants' ratings were higher for higher stimulus intensity levels in both modalities (linear mixed-effects model: pain: $\beta = 0.633$, $SE = 0.028$, $t_{(44)} = 22.42$, $p < .001$; vision: $\beta = 0.867$, $SE = 0.018$, $t_{(44)} = 47.06$, $p < .001$).

### Expectation generation from distribution cues

In the expectation task, participants were presented with distribution cues with varying levels of mean (30, 40, 50, 60, or 70), standard deviation (5 or 12.5), and skewness (negative, positive, or symmetric). These three distributional properties

were independently manipulated across cues in a fully crossed design. Following each cue, participants rated their expectations regarding noxious and visual stimuli (stimuli were not delivered during this task). The task included 360 trials overall (six repetitions of each combination of modality, cue mean, cue variance and cue skewness). The effects of cue mean, variance, and skewness on reported ratings were tested simultaneously with mixed effects models (one per modality) including predictors for each distributional property and their interactions, and controlling for block and trial number within each block.

**Effects of cue mean on expectations.** Expectation ratings significantly assimilated to the cue mean in both modalities (linear mixed effects model, pain: $\beta = 0.901$, $SE = 0.015$, $t_{(57.7)} = 62.72$, $p < .001$; vision: $\beta = 0.9135$, $SE = 0.021$, $t_{(51.3)} = 43.25$, $p < .001$; Fig 2A).

**Effects of cue variance on expectations.** Previous studies have mostly focused on the effect of cue variance on perception rather than expectations. This is because the variance is thought to influence the precision (certainty) of the expectations, not their value. Therefore, cue variance should affect cue influences on perception, with more certain predictions (lower variance) having stronger effects. Here, expectation ratings were significantly higher for lower cue variance in pain trials ($\beta = -0.045$, $SE = 0.014$, $t_{(58.1)} = -3.16$, $p = .002$; Fig 2A), but not in visual trials ($\beta = 0.003$, $SE = 0.013$, $t_{(66)} = 0.25$, $p = .800$). In other words, participants expected more painful stimuli after more certain cues only for pain stimuli. This finding is inconsistent with accounts such as Bayesian predictive processing or uncertainty aversiveness. We further tested the interaction between the effects of cue mean and variance on subsequent ratings, which based on Bayesian predictive processing accounts is expected to be significant for perceptual, and not expectation, ratings. Indeed, the cue mean x variance interaction was not significant in either modality (pain: $\beta = -0.0002$, $SE = 0.006$, $t_{(7954)} = -0.026$, $p = .979$; vision: $\beta = 0.014$, $SE = 0.007$, $t_{(7954)} = 1.93$, $p = .054$).

**Effects of cue skewness on expectations.** We tested for the first time how the skewness of the cue distribution affects expectations. Extreme values biased participants' expectations, which were higher for positively skewed compared

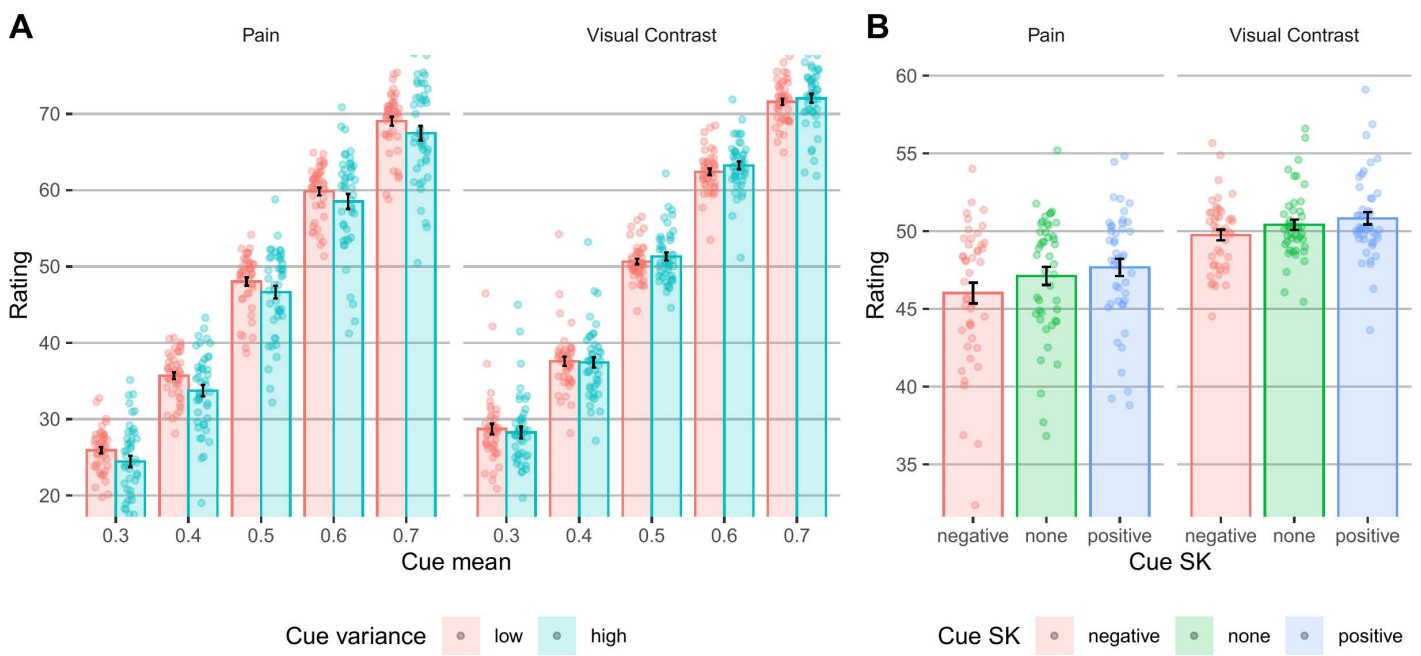

**Fig 2. Behavioral results - expectation task. (A)** Participants' expectation ratings as a function of the mean and variance of the cue's values, in pain and vision trials. **(B)** Participants' expectation ratings as a function of the skewness of the cue's values, in pain and vision trials. In both panels, bars represent the mean across participants, error bars represent the standard error of the mean across participants, and points represent single participants.

to symmetric cues (pain: $\beta = 0.032$, $SE = 0.009$, $t_{(7954)} = 3.59$, $p < .001$; visual perception: $\beta = 0.024$, $SE = 0.010$, $t_{(7954)} = 2.45$, $p = .014$; Fig 2B) and lower for negatively skewed compared to symmetric cues (pain: $\beta = -0.064$, $SE = 0.009$, $t_{(7954)} = -7.135$, $p < .001$; visual perception: $\beta = -0.037$, $SE = 0.010$, $t_{(7954)} = -3.77$, $p < .001$). These findings are consistent with over-weighting of extreme values when constructing expectations.

Overall, the effects of the skewness suggest that extreme values bias expectations, and the finding that participants expected more painful stimuli following more certain cues only in pain might suggest that participants were particularly attentive to lower extreme values in the context of pain.

**Computational model: weighting of cue values.** To more directly test whether participants weight all cue values equally or overweight particular values (e.g., inliers vs. outliers, or smaller vs. larger values), we developed a computational model of cue-based expectation generation. The model assumes that in each cue, each value is weighted based on its relative location in the distribution of the 10 values (Fig 3A). Each value's weight is based on a combination of a power term modeling the weighting of inliers vs. outliers with the free parameter $k$, and a separate logistic term modeling the weighting of values that are smaller vs. larger than the mean with the free parameter $b$ (see Methods for a detailed description). The model was inspired by the model of Spitzer et al. in the context of numeric estimation [50].

We fit the model to the expectation data to estimate the two free parameters, $k$ and $b$, for each modality and participant. The expectation model fit the data well overall, with an average correlation between predicted and empirical expectation ratings across participants of Pearson's $r = 0.945$ ($SD = 0.032$) for pain and $r = 0.928$ ($SD = 0.074$) for vision, and an average root mean squared error (RMSE) of 5.904 ($SD = 1.668$) for pain and 6.415 ($SD = 2.545$) for vision (on a rating scale of 0–100). The Pearson's correlation was lower, but still high and significant, when tested separately within each level of cue mean (range of mean correlation across participants, across all cue mean levels: pain 0.344-0.424; visual perception 0.188-0.375, all $p$s < .001). This indicates that participants tracked variation in cues accurately, and that the model predicted their responses well and performed generally better in pain than in vision (Fig 3B).

The free parameter $k$ measures over-weighting of inliers ($k < 1$) or over-weighting of outliers ($k > 1$), while $b$ measures over-weighting of values that are smaller ($b < 0$) or larger ($b > 0$) than the mean value. We found that participants significantly over-weighted outliers in both modalities (i.e., $k > 1$; Wilcoxon signed rank test; pain: median $k = 1.66$, $p = .019$; visual perception: median $k = 1.47$, $p = .008$), and also smaller values specifically in pain and not in visual perception (i.e., $b < 0$; Wilcoxon signed rank test; pain: median $b = -1.64$, $p < .001$; visual perception: median $b = 0.16$, $p = .446$) (Fig 3C). Furthermore, $k$ values, but not $b$ values, were correlated between the two modalities across participants (Spearman correlation; $k$: $\rho = 0.45$, $p = .002$; $b$: $\rho = 0.16$, $p = .304$).

Finally, we tested whether the estimated $k$ and $b$ values correlated with state/ trait scores based on questionnaires completed by the participants prior to the cued-perception task. For example, people with higher fear of pain, pain catastrophizing, or anxiety may focus more on cues predicting higher expected pain. However, no correlations between fear/ anxiety-related measures (Fear of Pain score [68], Pain Catastrophizing score [69], and State-Trait Anxiety Inventory score [70]) and optimized $k$ or $b$ were significant for either modality ($N = 45$; all $|r| <= 0.22$ and all $p >= .140$; see Table A in S1 Text).

Taken together, the behavioral and computational results of the expectation task suggest that while expectations strongly assimilate towards the mean cue value (Fig 2A), participants overweight extreme cue values in both modalities (Figs 2B and 3), and demonstrate an optimism bias specifically in pain, relying more on low than high extreme values (Fig 3). Such effects were not considered in most previous studies, but contrary to our hypothesis, they do not explain previous findings of uncertainty aversiveness in pain based on unintentional manipulation of extreme values when manipulating variance, since participants (at least in our sample) were more attentive to lower rather than higher extreme pain values.

## The effects of cue-based expectations on perception

In the cued-perception task (for a detailed description see Methods and Fig 1), participants viewed cues consisting of a distribution of 10 values (with a fixed mean value of 30 or 70, low or high variance, and negative, symmetric or positive

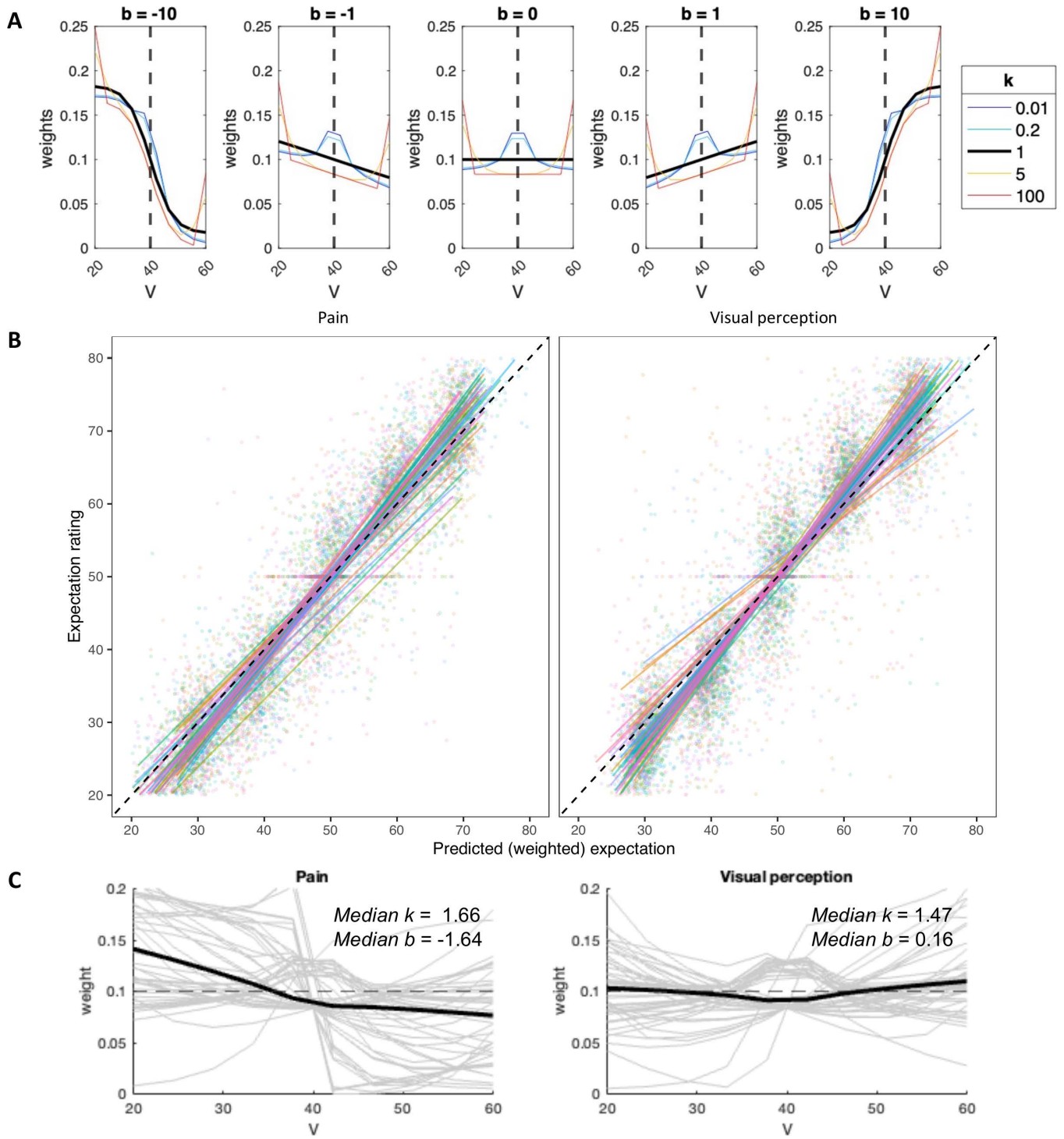

**Fig 3. Computational model of cue-based expectation generation. (A)** Simulations of the expectation model: Mapping of cue values, $V$ (10 per cue), to weights for expectation computation, based on the two free parameters of the model: $k$ and $b$. When $k = 1$ (black line), inliers and outliers are equally weighted. When $k < 1$ (cold colors), inliers are over-weighted, and when $k > 1$ (hot colors), outliers are over-weighted. When $b < 0$ (left panels), values below the mean are over-weighted, when $b > 0$ (right panels) values above the mean are over-weighted, and when $b = 0$ (middle panel), values are equally weighted. The dashed line represents the cue mean. **(B)** Correlations between the observed and predicted (based on the computational model) expectation ratings were very high across participants. Each line represents a single participant, and each dot represents a single trial. Data from

different participants are presented in different colors. **(C)** The weight function for each modality, based on the median *k* and *b* values across the group. Weight functions of individual participants are overlaid with thin, gray lines. The dashed line represents equal rating of all cue values (*V*). Note that panels A and C are based on a symmetric cue consisting of equally distributed values between 20 and 60.

skewness) followed by painful heat (47°C or 48°C) or flickering checkerboards (50% or 60% luminance contrast). This task included 144 trials overall, with three repetitions of each combination of modality, stimulus intensity, cue mean, cue variance and cue skewness. The effects of stimulus intensity and cue mean, variance, and skewness on reported ratings were tested simultaneously with mixed effects models (one per modality) including these predictors and their interactions, as well as block number and trial number within block as additional numeric regressors, to control for temporal effects such as habituation and sensitization [71].

As expected, participants rated stimuli with higher intensity levels (higher temperature or visual contrast) as more intense (pain: $\beta = 0.330$, $SE = 0.027$, $t_{(128.7)} = 12.40$, $p < .001$; visual perception: $\beta = 0.334$, $SE = 0.031$, $t_{(88.15)} = 10.630$, $p < .001$).

**Effects of cue mean on perception.** Participants' stimuli ratings assimilated to the cue mean (Fig 4A; pain: $\beta = 0.235$, $SE = 0.035$, $t_{(79.33)} = 6.633$, $p < .001$; visual perception: $\beta = 0.289$, $SE = 0.038$, $t_{(69.15)} = 7.521$, $p < .001$), replicating previous studies. Cue mean effects on pain and visual perception were correlated across participants (Pearson's r = 0.82, $t_{41} = 9.174$, 95% CI = [0.689, 0.899], $p < .001$), indicating that participants who were affected by the cues were similarly affected in both modalities.

**Effects of cue variance on perception.** The cue variance affected perception of noxious stimuli, such that higher cue variance led to higher ratings (pain: $\beta = 0.046$, $SE = 0.022$, $t_{(2845)} = 2.083$, $p = .037$), but there was no effect of the cue variance on ratings of visual stimuli ($\beta = 0.031$, $SE = 0.021$, $t_{(2875)} = 1.455$, $p = .146$). These results support the hypothesis that uncertainty is aversive particularly in the context of harmful stimuli [7], although this finding did not replicate in previous direct replication [37] and meta-analysis [41]. These effects were, however, very small, and sensitive to analytical variability: without covariates for block and trial number (habituation/sensitization across time), the effect was significant in vision but not pain, consistent with marginal effects in both modalities.

As described above, Bayesian predictive processing accounts predict that the effect of the cue mean (expectation value) on perception should be smaller with higher cue variance (lower expectation precision). In line with this prediction, when rating visual stimuli, the effect of the cue mean was smaller when the cue variance was higher (cue mean *x* variance interaction: $\beta = -0.056$, $SE = 0.021$, $t_{(2876)} = -2.673$, $p = .008$), but this interaction was not significant in pain ($\beta = -0.036$, $SE = 0.022$, $t_{(2849)} = -1.669$, $p = .095$).

**Effects of cue skewness on perception.** The skewness only affected visual perception (higher ratings after positively skewed vs. symmetric cues; $\beta = 0.064$, $SE = 0.030$, $t_{(2876)} = 2.147$, $p = .032$; no significant effect for negatively skewed vs. symmetric cues, $\beta = -0.009$, $SE = 0.030$, $t_{(2875)} = -0.307$, $p = .759$) and not pain (negative vs. symmetric: $\beta = -0.056$, $SE = 0.031$, $t_{(2845)} = -1.810$, $p = .070$; positive vs. symmetric: $\beta = 0.045$, $SE = 0.031$, $t_{(2845)} = 1.459$, $p = .145$) (Fig 4B).

Overall, ratings were higher following cues with higher mean in both modalities (Fig 4A). We also revealed a small and unstable effect of uncertainty aversion in pain, somewhat consistent with a previous study suggesting that uncertainty enhances threat and thus pain, but not with its failed replication and a recent meta-analysis. In visual perception, the effect of the cue mean was stronger for more certain cues, in line with predictive processing accounts. However, predictive processing accounts do not explain why the effect of the cue mean was not modulated by cue variance in pain, and why pain ratings were higher following higher uncertainty. Finally, the skewness only weakly affected visual perception, with more positive skewness leading to higher visual ratings. In addition, computational modeling indicated that perception assimilates towards expected values more strongly in pain compared to visual perception, in line with the hypothesis that pain perception is more sensitive to contextual information (see Results in S1 Text, section "Computational modeling of perception", comparison between $w_p$ and $w_v$).

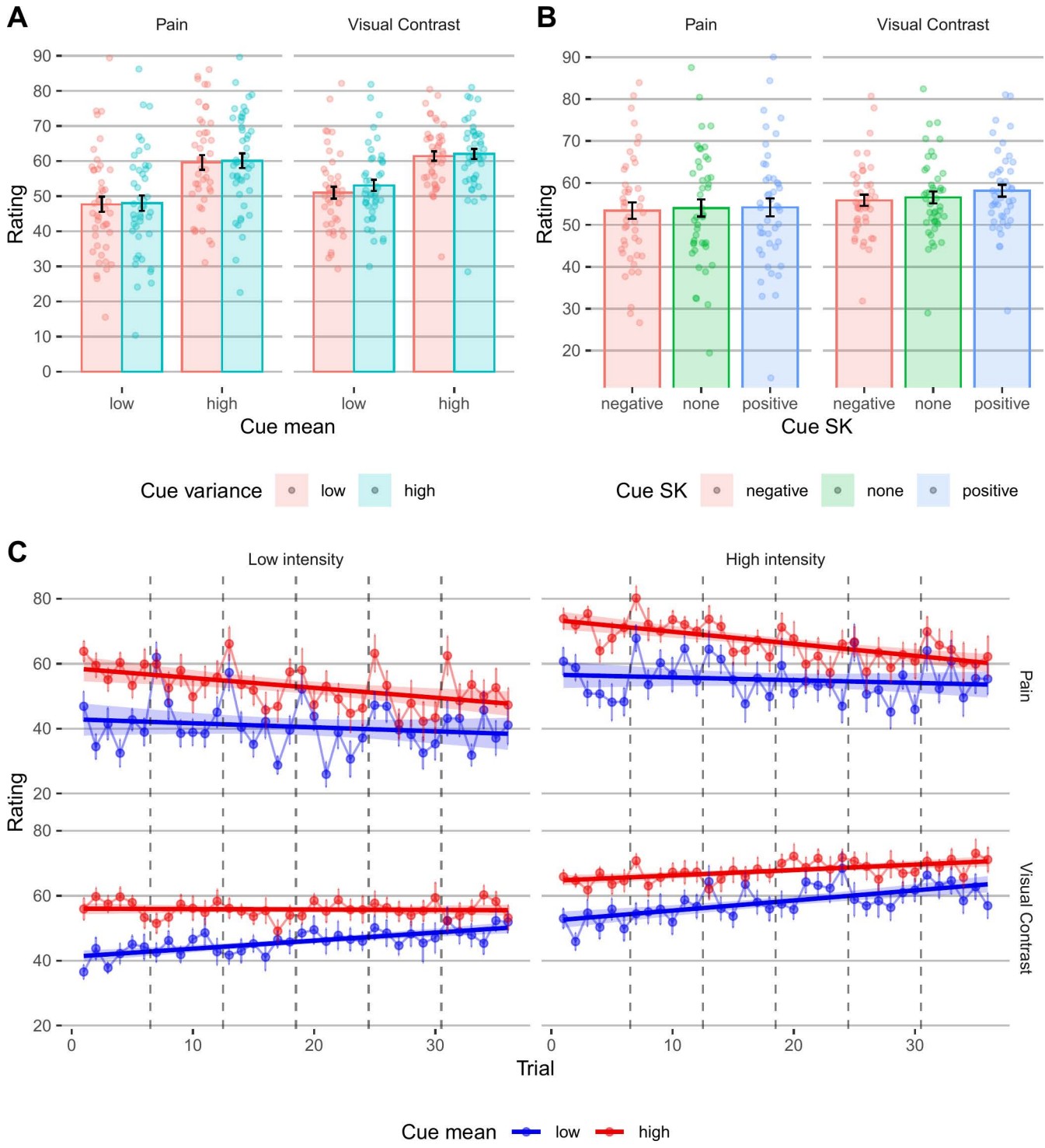

**Fig 4. Behavioral results, cued-perception task. (A)** Participants' perception ratings as a function of the mean (x axis) and variance (color) of the cue's values, in pain and visual trials (column). **(B)** Participants' perception ratings as a function of the skewness (x axis and color) of the cue's values, in pain and visual trials (columns). In panels A and B, bars represent means across participants, error bars represent the standard error of the mean across participants, and points represent single participants. **(C)** The effect of cue mean (color) on pain and visual contrast ratings over time, for low vs. high stimulus intensity (columns), in pain and visual perception (rows). Error bars represent standard error of the mean across participants. The lines represent the linear fit, and the shading represents 95% CIs of the linear fit. Vertical dashed lines separate between different blocks (note that a new skin site was used for each block, and thus the increase in the averaged pain rating for the first trial of each block stems from site-nonspecific sensitization and site-specific habituation [71]).

**Does the effect of cue-based expectation persist?** In the models described above for the effects of cue-based expectations on perception, we found a significant block effect in both modalities (pain: $\beta = -0.089$, $SE = 0.013$, $t_{(2848)} = -6.751$, $p < .001$; visual perception: $\beta = 0.096$, $SE = 0.013$, $t_{(2879)} = 7.495$, $p < .001$), and a significant trial effect in pain ($\beta = -0.140$, $SE = 0.013$, $t_{(2890)} = -10.97$, $p < .001$), but not visual trials ($\beta = 0.007$, $SE = 0.012$, $t_{(2905)} = 0.532$, $p = .595$). We tested whether the effect of the cue mean on perception persists throughout the task with mixed-effects models predicting the rating based on cue mean, trial number (modelled per modality continuously across the task, and not within each block) and their interaction, for each modality. The interaction between trial number and cue mean effect was significant in both modalities (Fig 4C; pain: $\beta = -0.055$, $SE = 0.014$, $t_{(2931)} = -3.904$, $p < .001$; visual perception: $\beta = -0.064$, $SE = 0.014$, $t_{(2953)} = -4.532$, $p < .001$), indicating that the effect of the cues decreased during the task. However, the cue mean effect was significant on the last block in both modalities (pain: $\beta = 0.188$, $SE = 0.030$, $t_{(41.8)} = 6.314$, $p < .001$; visual perception: $\beta = 0.188$, $SE = 0.045$, $t_{(43.7)} = 4.166$, $p < .001$), demonstrating a persistent cue effect. Computational modeling strengthened the conclusion that most participants did not learn to ignore the cues during the task, since a comparison between five different models indicated a better fit of models without learning compared to models with learning (see Results in S1 Text, section "Computational modeling of perception"), in line with previous studies [9,63,65,72].

**"Boundary effects" of cue-based expectations.** Previous studies have demonstrated that the effect of cue-based expectations on pain perception decreases when the difference between the sensory input and the expectation is too large [73]. In the current study, we have used fixed stimulus intensities and cue mean values (i.e., these values were not calibrated to each participant), which could potentially result in large discrepancies for some participants, inducing such boundary effects. To test for such effects, we performed additional analyses similar to those of Hird et al. [73], testing cubic and quadratic effects of the cue PE (the difference between the subjective stimulus value, based on the average rating per participant, modality and intensity from the stimulus-response task, and the cue mean) on the experienced PE (the difference between the reported rating and the subjective stimulus value; for additional details see Methods). In both modalities, the model revealed a significant effect of the cue PE on experienced PE (pain: $b = -0.460$, $SE = 0.056$, $t_{(2955)} = -8.142$, $p < .001$; visual perception: $b = -0.282$, $SE = 0.043$, $t_{(2981)} = -6.574$, $p < .001$), replicating our finding of a significant assimilation of perceptual ratings towards the cue mean value. However, there were no significant cubic or quadratic effects in both modalities (pain: cubic $t_{(2958)} = 1.120$, $p = .263$, quadratic $t_{(2961)} = 1.032$, $p = .302$; visual perception: cubic $t_{(2996)} = 0.157$, $p = .876$, quadratic $t_{(2996)} = -1.225$, $p = .221$), providing evidence against the presence of boundary effects in our data (Fig A in S1 Text).

## Cue and stimulus intensity effects on neural processing

**Effects on pain neuromarkers.** We focused on two *a priori* neuromarkers for pain (Fig 5A): (1) The Neurologic Pain Signature (NPS [74]), which is sensitive and specific to nociceptive pain across studies, tracks the intensity of nociceptive input, and predicts pain ratings with very large effect sizes in >50 published study cohorts [56,75]. (2) The Stimulus Intensity Independent Pain Signature (SIIPS [76]), which captures higher-level, endogenous influences on pain construction independent of stimulus intensity and the NPS score. In most previous studies, the NPS was not modulated by expectations or other contextual effects [62,77], although some previous studies have found modulations with some types of interventions [65,78], sometimes with very small effect size [56]. Conversely, the SIIPS was found to be affected by expectations and related psychological manipulations such as placebo treatment [62], perceived control and conditioned cues [77]. Notably, a study using social distribution cues like the ones we have used here found no effects of the cues on the NPS and SIIPS scores [10].

We computed a dot product-based score, the 'pattern response', for the stimulus period of each trial, and included the scores for each neuromarker in mixed-effects models identical to the ones used for the behavioral pain ratings. The NPS score was significantly higher for high compared to low intensity painful stimuli ($\beta = 0.143$, $SE = 0.027$, $t_{(2895.21)} = 5.23$, $p < .001$), replicating previous studies. However, it was not significantly affected by the cue mean ($\beta = 0.022$, $SE = 0.030$,

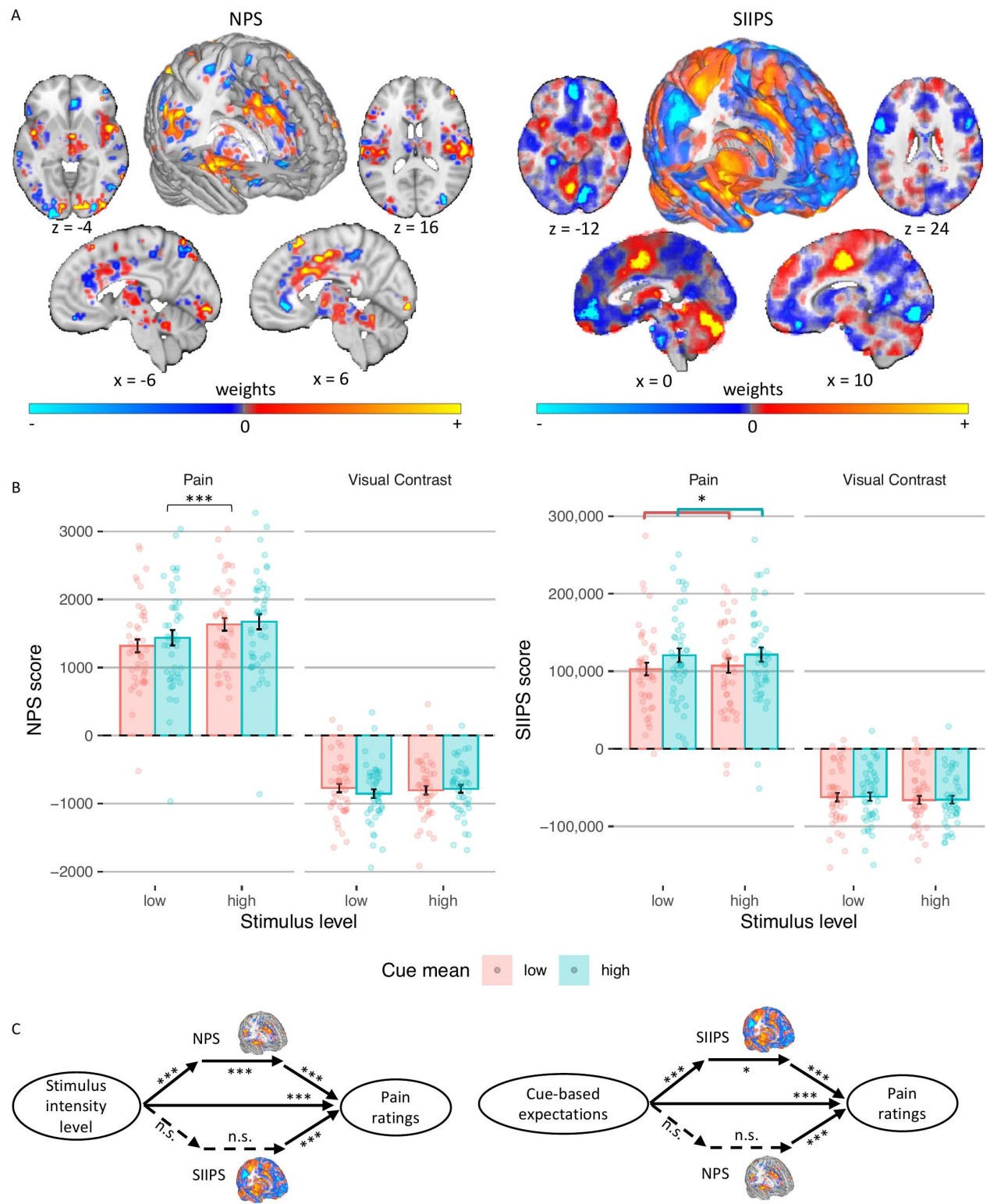

**Fig 5. Neuromarker results. (A)** The NPS and SIIPS neuromarkers. Adapted from Botvinik-Nezer et al. (2024), Nature Communications (DOI: https://doi.org/10.1038/s41467-024-50103-8) [62], under CC BY 4.0. **(B)** NPS (left) and SIIPS (right) score as a function of the stimulus intensity (x axis), cue mean (color) and modality (column). Bars represent means across participants, error bars represent the standard error of the mean across participants,

and points represent single participants. **(C)** Multilevel mediation analysis with pain neuromarkers. Solid lines represent significant effects and dashed lines represent non-significant effects. Asterisks represent the level of significance (* $p<.05$, *** $p<.001$).

$t_{(202.09)}$ = 0.74, $p=.462$), variance ($\beta=0.006$, $SE=0.027$, $t_{(2895.26)}$ = 0.24, $p=.812$), or skewness (positive vs. symmetric: $\beta=0.017$, $SE=0.038$, $t_{(2895.6)}$ = 0.45, $p=.653$; symmetric vs. negative: $\beta=0.047$, $SE=0.039$, $t_{(2894.92)}$ = 1.21, $p=.225$). The cue mean x cue variance interaction was also not significant ($\beta=0.006$, $SE=0.027$, $t_{(2904.36)}$ = 0.21, $p=.836$). Thus, the NPS response depends on the noxious input, and is not modulated by the cues (Fig 5B).

Conversely, the SIIPS score assimilated towards the cue mean ($\beta=0.066$, $SE=0.028$, $t_{(2936.51)}$ = 2.33, $p=.020$; (Fig 5B)), but was not significantly affected by the stimulus intensity ($\beta=−0.017$, $SE=0.028$, $t_{(2936.19)}$ = -0.59, $p=.552$), cue variance ($\beta=0.016$, $SE=0.028$, $t_{(2937.62)}$ = 0.55, $p=.586$), or cue skewness (positive vs. symmetric: $\beta=−0.012$, $SE=0.040$, $t_{(2937.06)}$ = −0.31, $p=.757$; symmetric vs. negative: $\beta=−0.011$, $SE=0.040$, $t_{(2936.81)}$ = −0.27, $p=.785$). The cue mean x cue variance interaction was not significant ($\beta=−0.007$, $SE=0.028$, $t_{(2936.44)}$ = −0.24, $p=.814$) as well. Importantly, the NPS and SIIPS are pain neuromarkers, and should not respond to neutral visual stimuli. Indeed, when testing the NPS and SIIPS scores during perception of visual stimuli, there were no significant effects of stimulus intensity or cues (all $p$s ≥ .094; Table C in S1 Text).

**Multilevel mediation analysis.** We used mediation analysis to test whether the NPS and/ or SIIPS formally mediate the effect of the cue-based expectations and/or stimulus intensity on trial-by-trial pain reports (Fig 5C). Trial-by-trial expectancy scores were based on the computational model (see "Computational model: weighting of cue values" above). Expectancy models controlled for stimulus intensity as a covariate, and vice versa.

The NPS partially mediated the effect of stimulus intensity level on pain ratings (*path ab* stimulus intensity level→NPS score→pain rating: $\beta=0.02$, $SE=0.00$, $z=3.68$, $p<.001$), controlling for cue-based expectations. Higher stimulus intensities led to higher NPS scores (*path a*, $\beta=0.11$, $SE=0.01$, $z=4.00$, $p<.001$), and higher NPS scores predicted greater trial-by-trial pain (*path b*, $\beta=0.28$, $SE=0.02$, $z=3.85$, $p<.001$). However, the NPS did not mediate the effect of the cue-based expectations on pain ratings (*path ab* cue-based expectation→NPS score→pain rating: $\beta=0.00$, $SE=0.00$, $z=1.42$, $p=.154$), controlling for stimulus intensity level, since higher cue-based expectations did not lead to higher NPS scores (*path a*, $\beta=0.02$, $SE=0.02$, $z=1.24$, $p=.216$).

Conversely, the SIIPS partially mediated the effect of the cue-based expectation on pain ratings (*path ab* cue-based expectation→SIIPS score→pain rating: $\beta=0.003$, $SE=0.001$, $z=2.45$, $p=.014$), controlling for stimulus intensity level. Higher cue-based expectations led to higher SIIPS scores (*path a,* $\beta=0.06$, $SE=0.01$, $z=3.88$, $p<.001$), and higher SIIPS scores predicted greater trial-by-trial pain (*path b* SIIPS score→pain rating: $\beta=0.15$, $SE=0.03$, $z=3.82$, $p<.001$). The SIIPS did not mediate the effect of the stimulus intensity level on pain ratings (*path ab* stimulus intensity level→SIIPS score→pain rating: $\beta=0.00$, $SE=0.00$, $z=1.37$, $p=.171$), controlling for cue-based expectations, since stimulus intensity was not associated with SIIPS scores (*path a*, $\beta=0.02$, $SE=0.01$, $z=1.23$, $p=.219$).

As expected, neither neuromarker mediated the effect of the stimulus intensity level or the effect of the cue-based expectation on the contrast rating of visual stimuli. Overall, the NPS score was only affected by the heat intensity and formally partially mediated the effect of the stimulus intensity on pain ratings, while the SIIPS score assimilated to the cue mean and partially mediated the effect of the cue-based expectations on pain ratings. Other properties of the cue, including the cue variance, skewness, and the interaction between the cue mean and cue variance, did not affect either neuromarker. These results indicate that the cues did not affect pain via changes in early nociceptive pain processing, but rather via systems associated with endogenous contributions to pain perception [62].

**Effects on ROIs related to early neural perceptual processing.** To complement neuromarker analyses, which capture activity in distributed systems, we tested whether the cues affected neural responses to the stimuli in several a priori ROIs related to pain and visual perception (Fig 6; for all regions see Table B in S1 Text). We first focused on regions

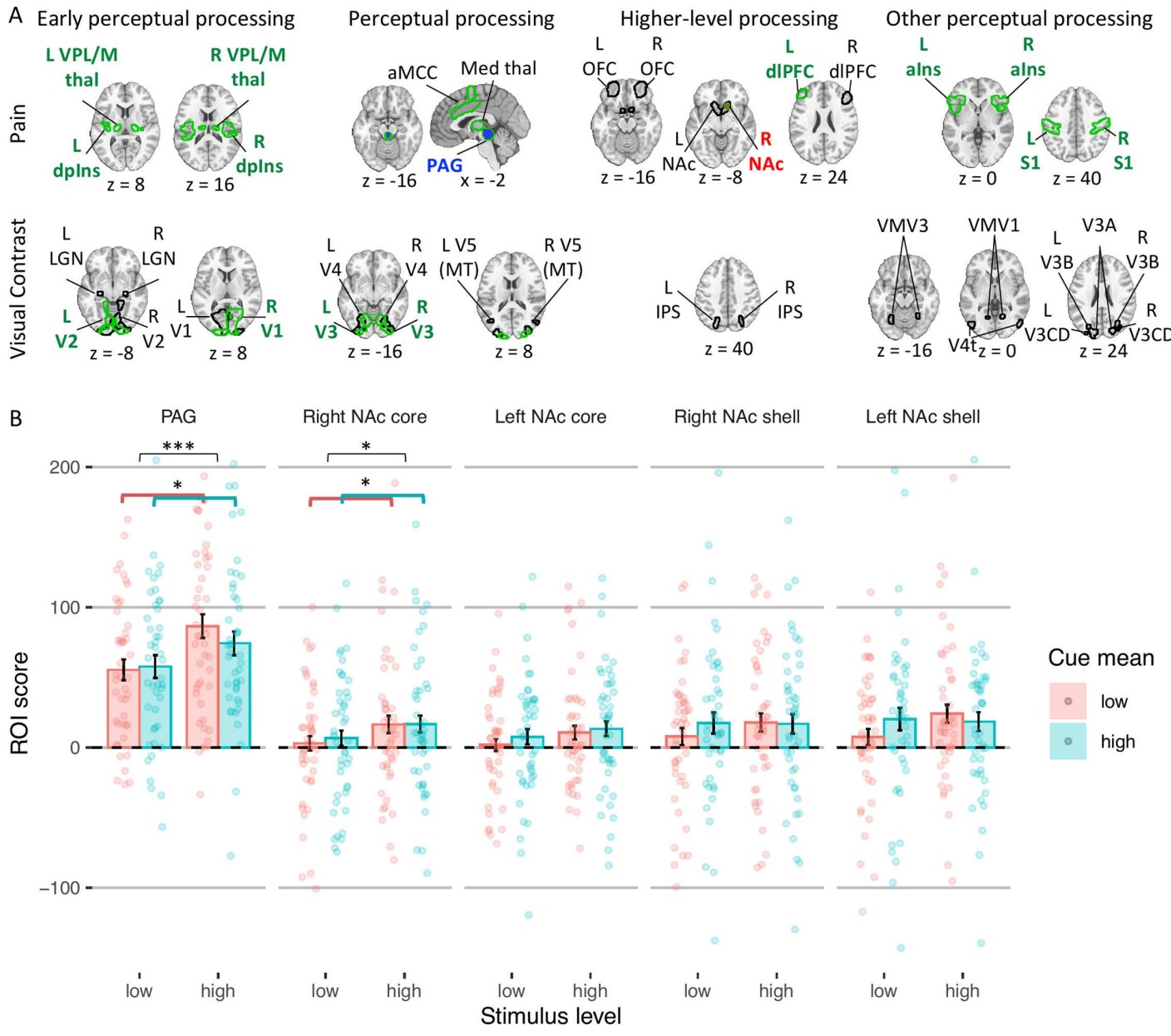

**Fig 6. ROI results. (A)** ROIs are shown with a contour for each modality (rows) and processing stage (columns). Regions with higher activity for more intense stimuli are presented with a green contour. Regions with a significant cue mean effect are presented in red (higher activity for higher cue mean) or blue (higher activity for lower cue mean). **(B)** The ROI score in the two regions with significant cue mean effect during pain perception (including other parts of the NAc) as a function of the stimulus intensity level (x axis) and cue mean (color). Bars represent means across participants, error bars represent standard error of the mean across participants, and points represent single participants. Asterisks represent the level of significance (* $p < .05$, *** $p < .001$). Abbreviations: aIns = anterior insula; aMCC = anterior midcingulate cortex; dlPFC = dorsolateral prefrontal cortex; dpIns = dorsal posterior insula; L = left; Med = medial; NAc = nucleus accumbens; OFC = orbitofrontal cortex; PAG = periaqueductal gray; R = right; thal = thalamus.

associated with early perceptual representations (for full statistics see Tables D and E in S1 Text). For nociception, we focused on the spinothalamic tract, including ventral posterior (VPL/VPM) thalamus and dorsal posterior insula (dpIns). Activity in these regions was larger for more intense heat stimuli (all $ps \leq .003$), but was not affected by the cue mean, cue

variance, cue skewness, or cue mean x cue variance interaction (all $p$s ≥ 0.141). For early visual perception, we focused on the lateral geniculate nucleus (LGN) and primary and secondary visual cortex (V1 and V2). Activity was higher for higher visual contrast in the right V1 ($p$ = .041, only uncorrected, i.e., this result does not survive correction for multiple comparisons [see Methods]) and left V2 ($p$ = .028, only uncorrected). There was no effect of stimulus intensity on activity in the LGN (right $p$ = .438; left $p$ = .332), left V1 ($p$ = .067) and right V2 ($p$ = .058). Like early nociceptive pain perception, early visual perception was not affected by the cues (negative vs. symmetric cues in right V2 $p$ = .072 and left V2 $p$ = .056; all other $p$s > 0.153). Thus, in sum, most of these areas showed stimulus intensity effects, but none were influenced by the cues.

**Effects on ROIs related to neural perceptual processing.** We then tested cue effects on regions associated with higher-level perceptual processing and affect (Fig 6; see Tables F and G in S1 Text for full statistics). For pain, these included the periaqueductal gray (PAG), anterior midcingulate cortex (aMCC), and medial thalamus. For visual perception they included V3, V4 and V5 (MT). Activity in all pain processing regions was again higher for higher noxious stimulus intensity (all $p$s ≤ .001), but only the PAG was sensitive to predictive cues: PAG activity was higher following low- vs. high-mean cues ($p$ = .031, only uncorrected), consistent with aversive PE-like responses found in the PAG in previous studies [79,80]. In the visual perception task, activity in visual regions was only higher for higher visual contrast in V3 (right $p$ = .046, left $p$ = .017, both only uncorrected; all other $p$s ≥ .193). While the cue mean, cue variance, and the interaction between them was not significant for any of these visual regions (all $p$s ≥ .106), the cues' skewness did affect activity: Activity was lower following negatively skewed compared to symmetric cues in bilateral V3 (right, $p$ = .009; left, $p$ = .032, both only uncorrected), bilateral V4 (right, $p$ = .010; left, $p$ = .045, both only uncorrected), and left V5 ($p$ = .027, only uncorrected). Activity was not different following positively skewed compared to symmetric cues in all regions (all $p$s ≥ .111). Overall, the results replicate the PAG's role in PEs, but provide no evidence for assimilation to cue value or encoding of cue precision in pain, and only weak evidence for assimilation to extreme low-value cues in visual areas.

**Effects on ROIs related to higher-level processing.** Next, we tested regions associated with higher-level affective and cognitive processing of noxious and visual stimuli, including the nucleus accumbens (NAc; shell-like and core-like parts), dorsolateral prefrontal cortex (dlPFC) and mid-lateral orbitofrontal cortex (OFC) for pain, and the intraparietal sulcus (IPS) for visual perception (Fig 6; see Tables H and I in S1 Text for full statistics). Activity was higher for more intense heat stimuli in the right NAc (core-like part; $p$ = .017, only uncorrected) and left dlPFC ($p$ = .001). The cues only affected activity in the NAc (core-like part; all other $p$s ≥ .070): Activity in the right NAc was higher following cues with a higher mean ($p$ = .018, only uncorrected) and lower variance (i.e., more certain; $p$ = .011, only uncorrected), and activity in the left NAc was higher for low-variance cues ($p$ = .001) and showed a cue mean x cue variance interaction ($p$ = .015, only uncorrected), such that cue mean effects were smaller for more precise cues. These findings were not in line with previous predictive coding findings [25,81] (see Discussion). In visual perception, activity in the IPS was not affected by the stimulus intensity or the cues (cue mean effect: $p$ = .066 in the left and $p$ = .074 in the right IPS; cue mean x cue variance interaction in the right IPS $p$ = .064; all other $p$s ≥ .216).

**Effects on other perceptual processing regions.** Finally, we performed an exploratory analysis testing a larger set of regions, including anterior insula (aIns) and primary somatosensory cortex (S1, mostly the hand area) for pain processing, and V3A, V3B, V3CD, V4t, VMV1, VMV2, and VMV3 for visual processing (Fig 6; see Tables J and K in S1 Text for full statistics). Activity during pain perception was higher for higher stimulus intensity in the aIns (right $p$ = .007; left $p$ = .001) and S1 (both $p$ < .001), but the cues did not affect their activity (negatively skewed vs. symmetric cues in the right aIns $p$ = .075, all other $p$s ≥ 0.225). Interestingly, activity was higher (less negative) with more intense heat in several visual regions, possibly because noxious heat stimuli produce diffuse effects on neuromodulatory systems, but none showed cue effects (see Results in S1 Text, section "Effects on neural perceptual processing"). In response to visual stimuli, none of the additional regions were affected by the stimulus intensity or cues, except for V3A, where activity was lower following negatively skewed compared to symmetric cues (right $p$ = .021, left $p$ = .010, both only uncorrected).

Overall (see Figs B and C in S1 Text for all effects across all regions), strong effects of stimulus intensity in pain and moderate effects in visual perception validated the sensitivity of ROIs to stimulation, but cue effects on stimulus-evoked activity were limited. In pain, we found aversive PE-like responses in the PAG and, in NAc, assimilation to cues and effects of cue uncertainty. However, the former effects did not survive correction for multiple comparisons, and the latter effects were not in line with previous theoretical predictions and findings [25,81–83]. In visual perception, we found limited assimilation towards extreme low-value cues in some visual areas. Finally, several nociceptive regions, as well as the NPS, assimilated to the cue mean during the anticipation period, but were not affected by the cue's precision (see Results in S1 Text, section "Effects on anticipatory neural activity").

## Discussion

Current models of perception emphasize the importance of predictive processes in constructing perceptual experience across modalities. Perception has been shown to assimilate towards expected values in different modalities, such as pain [7,8,10–12], vision [5,13,14], audition [84–86], taste and olfaction [87], and interoceptive experiences like itch [88] and nausea [89]. Consistent with these findings, in the current study, reported perceptions showed strong assimilation towards predicted values (cue mean) in both pain and visual intensity judgments. Furthermore, this assimilation decreased but persisted throughout the experiment although the cues were not reinforced, and was highly correlated across modalities, suggesting domain-general susceptibility to cues and/or learning. While several brain regions responded more to more intense stimuli, only neural activity related to higher-level pain processing was modulated by the cue mean. Behaviorally, in line with Bayesian predictive coding accounts [5,6,30], we found that overall, the effect of the cue mean on visual perception was stronger when cue precision was higher, but this was not true for pain perception. Furthermore, pain ratings were overall higher following less precise cues, and no brain regions showed activation patterns consistent with the effects of the cue variance (but see below). Finally, we found that when generating expectations, people are strongly drawn towards the cue mean, but overweight extreme values across modalities, and also smaller values specifically in pain.

Our findings are partly consistent with Bayesian predictive processing accounts, but provide new evidence addressing several open questions. First, while the prediction regarding assimilation to expected values is straightforward at the behavioral level, where and how the brain encodes this information is much less clear. One hypothesis is that brain representations related to perceptual experience should assimilate towards predicted values. Here, early nociceptive and visual regions – and a neuromarker for nociceptive pain, the NPS – did not show evidence for assimilation towards predicted values during perception, nor did higher-level perceptual regions such as the aMCC and aIns in pain or V3, V4 and V5 in visual perception. Only the right NAc and SIIPS, a distributed neuromarker for pain related to endogenous sources (including signal in dmPFC, aIns, vmPFC, and other regions), showed significant (but relatively weak) assimilation to predicted values. These results suggest that early perceptual processes are relatively shielded from influences of conceptually driven predictions, and the strong effects on perception are driven by higher-level evaluative processes, in line with psychological accounts questioning early influences of context information on perception [58] and recent findings in the context of placebo analgesia [62]. Such effects might also depend on the type of cues used. Indeed, a previous study using cues similar to those we used here has found that informational cue effects were mediated by frontoparietal areas involved in high-level construction of perception and value (e.g., dlPFC, OFC and IPS), whereas classically conditioned cue effects were mediated by subcortical systems [10].

Predictive processing theories also emphasize coding of PEs ('predictive coding' [6,67,90]), a contrast with predictions that provide efficient representations of salient changes in the environment and drive the updating of internal models (learning) [32,82,91]. A previous study has demonstrated "boundary effects" on cue-based expectations, such that the effect of cues on perception was decreased when the actual sensory input deviates too much from expectations [73]. We did not find evidence for such boundary effects in our data, although notably we did not systematically vary the PE, and

the subjective stimulus values used to compute the PE were based on ratings provided in the stimulus-response task, which was performed on a different day from the cued-perception task.

Which areas show brain responses that assimilate to predictions or contrast with them (encode PE) is currently an active area of investigation. A working hypothesis is that early sensory areas encode sensory PEs, as they are relatively distal from internal models and shielded from their effects, whereas higher-level perceptual areas encode posterior perceptions, which represent stimuli but assimilate to predicted values. This may be particularly true when internal predictive models are conceptual, including the kinds of non-reinforced cues about others' experience that we studied here. Here, PAG activity was higher following low vs. high cue mean, consistent with pain-related aversive PE that was found in the PAG in previous studies [79,80] (but this result did not survive correction for multiple comparisons). Together with the assimilation towards predicted values in higher-level systems, our results largely support this working hypothesis, although we do not directly test encoding of different types of PEs [90].

Bayesian accounts of brain function also make specific predictions about the precision of expectations. More certain (precise) cues should have stronger effects on perception [5,6,30]. Some previous studies have provided support for this hypothesis by using the precision of stimulus history as a proxy for predictive precision [25,34]. The kinds of cues with distributions we used here offer particular advantages in studying cue precision, as the cue mean, variance (i.e., inverse precision) and other properties (e.g., extreme values) can be independently experimentally manipulated across cues. Studies using this type of cues have found results that contradict predictive processing accounts by finding null effects of cue precision [37] or even direct effects of predictive uncertainty on increased pain perception [7]. In the current study, cue precision effects supported Bayesian predictive processing accounts in the visual modality, by showing a cue mean x variance interaction. In contrast, more uncertain cues led to higher ratings of painfulness of noxious, but not visual, stimuli, suggesting that uncertainty may be aversive in the context of pain, as previously suggested by Yoshida et al. [7]. However, these findings were previously not replicated [37] and not supported by a recent meta-analysis [41]. Here, the effect sizes were small and of marginal significance in pain, and not robust to methodological variations in the covariates included.

Several studies have recently explored the neural correlates of expectations' precision. Predictive processing accounts predict that perceptual activity will be higher following less precise cues, because of stronger reliance on incoming sensory information. Indeed, studies have shown decreased PAG fMRI activity [25] and early EEG responses [81] for more precise expectations. Here, perceptual responses were not affected by the cue precision in both modalities, including in the PAG. The only region that was affected by the cue precision was the NAc, where activity was higher following more (rather than less) precise cues during pain perception. While this effect was in opposite direction from the effects previously shown in other (earlier) regions, it could be consistent with predictive processing, since activity in the NAc might represent the priors (expectations) and thus is expected to be stronger when the priors are more precise (and also when they are stronger, and indeed NAc activity assimilated to the cue mean).

The persistent effects of the non-reinforced cues in both pain and visual perception might represent modulation of early perceptual processes, modulation of perceptual decision-making (higher-level affective and cognitive processes), or a post-experience report bias (i.e., demand characteristics [92], or conformity to alleged ratings of other participants [93]). Our neural results suggest that cues affect perceptual decision-making, since activity in regions related to early perceptual processing was not affected, while neural activity related to higher-level processing was affected by the cues, at least in pain. Moreover, neural effects were different between the two modalities. This could indicate either that expectation effects are modality-specific rather than modality-general, even when they operate on higher-level processes, or that there were other meaningful differences between the two modalities in our experiment, such as difference in stimulus ambiguity.

Beyond the effect of the cues on perception, our design also allowed us to study how expectations are generated from multiple values, and anticipatory neural responses. Reported expectations assimilated towards mean values. Anticipatory responses in several pain regions, including the insula, aMCC, PAG, and even the NPS, were higher for high vs. low cue mean, while the opposite direction was found on visual trials, with lower anticipatory activity for high vs. low cue mean

in several visual regions (Results in S1 Text). We also tested the effects of extreme cue values, and how participants weight predictions of upcoming pain and visual contrast across the distribution from low to high values. Such weighting has been studied in the context of magnitude judgments [49,50] but not, to our knowledge, in pain. We hypothesized that over-weighting of extreme values might account for previous inconsistent findings in the context of the effect of cue variance on pain, if participants over-weight extreme high-pain values (e.g., attend most to the most threatening potential values) and extreme values were unintentionally manipulated with the variance. However, this hypothesis was not supported by our data, since (1) participants in our sample were more attentive to low, rather than high, pain values when generating expectations; (2) higher cue variance led to slightly higher pain ratings even when outliers were orthogonal and controlled for; and (3) perceptual ratings were drawn towards larger extreme values only in visual (and not pain) perception. Notably, such effects of nonlinear weighting might depend on the characteristics of the sample. The current sample consisted of healthy young adults, who may have been looking for safety signals as part of an optimism bias, or may have viewed themselves as less sensitive to pain compared to others (and thus trusted lower ratings more). Putatively, different populations, such as chronic pain patients, might instead look for risk signals or view themselves as more sensitive to pain, and thus overweight larger values. Perhaps surprisingly, the tendency to overweight smaller values when forming expectations did not correlate with personality or state measures such as fear of pain, pain catastrophizing, or state anxiety. However, a growing literature suggests that task-based and self-report measures are often unrelated [94,95], and assess different constructs over different contexts and time scales [96,97].

There are several potential explanations for the discrepancy between cue effects on expectation ratings and perceptual judgments. Participants may rely on different cognitive strategies when explicitly rating their expectations versus when making perceptual judgments. When explicitly asked to report expectations, participants might engage in a more deliberate, model-based process that incorporates cue information differently than when making perceptual decisions, which could rely more on heuristics or sensory evidence. Prior research suggests that the act of making expectancy ratings itself can alter learning and decision-making processes, potentially by encouraging a more reflective or abstract representation of the cue-stimulus relationship [98]. This could mean that expectation ratings reflect an explicit cognitive model of the task, whereas perceptual judgments are more influenced by direct sensory experience and task demands. Another possibility is that the effects of variance or outliers in the cue distributions may be too subtle to influence perceptual judgments once expectation is weighted against stimulus input. The predictive influence of expectation might be diminished when it does not provide strong enough information to override perceptual uncertainty. Alternatively, participants may learn over time to downweight or ignore cues if they prove to be unreliable predictors of the actual stimulus, leading to reduced cue effects on perception even if they still influence explicit expectation ratings; however, our results indicate that the effect of the cue persisted, and thus this explanation is less likely. Further work is needed to disentangle whether this discrepancy arises from differences in cognitive strategies, cue weighting, or the extent to which expectations interact with sensory processing in a given task context.

Several limitations should be considered in the context of the current findings. First, we have used cues that were not predictive of actual stimulus intensity (i.e., non-reinforced), with fixed mean levels that did not depend on participants' individual psychometric curves (i.e., how intense the stimulus is expected to be perceived by the specific individual), and fixed stimulus intensity levels that were not calibrated to each participant. The use of non-reinforced cues was by design, as this allowed us to study the effects of conceptual information on brain and behavior, without influences of conditioning of nociceptive responses or other forms of associative learning. Though it is perhaps surprising that such "mere suggestions" without true information about the stimulus can have strong effects, such cues have been shown to yield robust behavioral effects that are comparable in size to those of cues that are predictive of actual intensity [10,63,64]. We also found similarly robust effects here, which decreased but persisted over the course of the experiment. The persistence of the effects over time here is also perhaps surprising based on the notion that even conditioned responses will generally extinguish if cues do not predict unconditioned stimuli in a valid way. However, this persistence has been replicated in multiple studies [10,63,65] and may be explained by a 'self-reinforcing' feedback loop detailed in Jepma et al. 2018 [65], whereby

initial expectations influence both stimulus processing and learning to create persistent effects of initial beliefs. Second, the effects of the cue variance and skewness that were found were almost all relatively weak, and the main driver of cue effects on perceptual processing was the cue mean. Third, the inclusion of both modalities in the same experiment with interleaved blocks may have yielded cross-modality dependency that could have driven some of the results, such as the correlation between the effect of the cue mean on pain and visual ratings. Finally, we did not directly compare the effects between pain and visual perception, since the scales are not directly comparable between modalities (e.g., a 10-point or a 20% increase in visual contrast rating is not necessarily perceptually equivalent to the same increase in pain rating).

Taken together, our findings suggest that perception is more complex than the recent Bayesian-driven focus on the mean and uncertainty of contextual information. More specifically, they show that extreme values have an important role in how people integrate information into expectations that later affect perception. Furthermore, these findings suggest that some aspects of expectation formation and their effect on subsequent perception are modality-general (e.g., the assimilation towards the cue mean and the importance of extreme values), while others are modality-dependent (e.g., the over-weighting of smaller vs. larger values). Better understanding of these processes and differences between sub-populations and modalities would advance our knowledge of how people form expectations, how these expectations affect perception, and how such processes could be leveraged to improve well-being and clinical care.

## Methods

### Ethics statement

The Institutional Review Board of the University of Colorado Boulder approved the protocol (approval number 16–0072) and participants provided a written informed consent before the beginning of the experiment.

### Participants

Forty-five healthy participants (25 females; age range 18–42, mean 24.1 years) completed the study. Five additional participants were excluded because they did not complete the experiment or reported impaired vision or recent substance abuse. Participants reported no history of neurological, psychiatric, or dermatologic conditions, and had not taken any medication during the 48 hours period prior to their participation. Participants reporting acute and chronic pain conditions in an initial online screening were excluded from participation. We recruited participants from the University of Colorado Boulder and local community through flyers and online ads.

### Procedure overview

Participants completed three different tasks and a set of questionnaires. First, the stimulus-response task served to familiarize participants with the stimuli and estimate the stimulus-response functions for heat pain stimuli of different temperatures and flickering checkerboards of different visual contrasts. Second, in the expectation task, participants (who were blinded to the design and manipulation) rated their expectations about heat pain and visual contrast intensities based on the cues provided in each trial without any actual stimulation. Third, in the cued-perception task, participants rated the perceived painfulness of noxious stimuli and contrast of visual stimuli that were preceded by the presentation of cues.

In session 1, the stimulus-response and expectation tasks were completed outside the MRI scanner in a behavioral laboratory. In session 2 (on average 4.4 days after session 1, range 1–11 days), participants first completed a battery of questionnaires on a computer, followed by a brief practice of the cued-perception task outside the scanner (including the cues and experiencing actual stimuli), which included six trials overall (three from each modality). Participants then moved to the MRI scanner where they completed another brief practice inside the scanner to get accustomed to the trackball and the MRI environment. Following a structural MRI scan, they completed the cued-perception task while their brain activity was measured with fMRI. After the cued-perception task, a functional localizer with different visual checkerboard contrasts was presented to the participants. Finally, participants completed a debriefing questionnaire outside the scanner.

PLOS Computational
Biology

## Stimulus-response task

In the first task, topical heat stimuli of five different temperatures and flickering checkerboards of five different luminance contrasts were presented to participants. After each trial, participants rated the perceived painfulness of the heat or luminance contrast of the checkerboard. The task was split into two blocks, one pain block and one vision block. In each block, 25 stimuli were presented in randomized order.

Temperatures in the heat block ranged from 45-49°C, in steps of 1°C. Each temperature was used five times and each stimulation lasted for two seconds (including ramp up and down from a 32°C baseline). After a variable delay of 5 – 6 seconds, participants rated the perceived heat intensity on a computerized visual analogue scale (VAS) by moving a vertical cursor bar on the VAS. The VAS was anchored at "no pain at all" and "worst pain imaginable". Participants were instructed to rate any heat stimulus that was noticed but not painful at the low extreme and that "worst pain imaginable" referred to the context of this experiment. They were further instructed to rate the maximum only in the case that they would have lifted the thermode to stop the experiment, which did not happen. After an inter-trial-interval (ITI) of 2 – 5 seconds, the next trial started. Before the task began, participants experienced two heat stimuli of 47°C without rating, to get familiar with the heat stimulation and reduce potential anxiety. Ratings were converted to a range between 0–100 in all tasks and conditions.

Black-and-white radial checkerboards were presented at 5%, 27.5%, 50%, 72.5%, and 95% luminance contrast. Checkerboards were presented for two seconds in each trial and the contrast of each checkerboard reversed at a frequency of 8 Hz to increase responses in visual brain regions. Checkerboards covered 12° visual angle in both the behavioral and the fMRI experiment and both screens were calibrated to display the same luminance level and contrast. After each stimulus, participants rated the perceived contrast on a VAS anchored "no contrast at all" and "strongest contrast". Similar to the pain block, an ITI of 2 – 5 seconds separated two consecutive trials.

## Expectation task

In the expectation task, participants saw different distributions of intensity ratings on a computer screen and reported their expected intensity of either heat pain or luminance contrast based on these ratings. Participants were informed that the rating distributions represented the ratings of other people for heat and checkerboard stimuli similar to those they just experienced in the stimulus-response task. The ratings were marked as vertical bars on visual analogues scales as used in the stimulus-response task (see Fig 1). A set of 10 ratings was presented on a single VAS in each trial for 1.5 seconds. After a brief delay of 1.75 seconds, participants rated their expected pain intensity or contrast intensity on the same VAS as in the stimulus-response task.

The distribution of ratings shown to the participants were determined by factorial combinations of their different distribution means (five levels, $M \in [30, 40, 50, 60, 70]$), standard deviations (two levels, $SD \in [5, 12.5]$), and skewness (three levels: negative, positive, or symmetric). Symmetric distributions were drawn from a normal distribution and skewed distributions were drawn from a log-normal distribution. Skewness was < -0.3 for negatively skewed, > 0.3 for positively skewed, and between -0.035 to 0.035 for symmetric distributions. Here, only 9 instead of 10 elements were sampled from the log-normal distribution. A 10th element was added between 2.0 – 2.5 standard deviations above or below the mean, respectively, to ensure that each skewed distribution included at least one extreme rating. All rating distributions were re-sampled if necessary to ensure that the properties of the distribution matched the specified mean, standard deviation, and skewness, as well as the independence of the factors across trials.

Participants completed a total of 360 trials in this task split into six blocks. Participants rated heat pain expectations in three consecutive blocks and contrast intensity in the other three consecutive blocks, with the order counterbalanced across participants (three pain and then three visual or three visual and then three pain blocks). The combination of the four factors (modality, mean, standard deviation, and skewness) resulted in a total of 60 combinations. Each combination was repeated six times. The order of trials was randomized for each participant within each block.

## Cued-perception task

The cued-perception task combined the two previous tasks into a single task performed inside the MRI scanner. In each trial, participants were cued with rating distributions (as in the expectation task) for 1.5 seconds before being presented with either cutaneous heat or a flickering checkerboard for two seconds (as in the stimulus-response task). Cues and stimulation periods were separated by a brief interval of 2.5 - 5.5 seconds. Participants rated the perceived intensity of the heat or luminance contrast after another brief interval of 3–6 seconds. Participants had 4.5 seconds to rate the intensity using an MRI compatible trackball. Trials were separated by an ITI of $3 - 6.5$ seconds.

In this task, stimulation modality (heat vs. visual) and two levels of stimulation intensity (47°C or 48°C, or 50% or 60% luminance contrast) were combined with different rating cue distributions similar to those used in the expectation task. The rating cues were drawn from distributions with two means ($M \epsilon [30,\ 70]$), two standard deviations ($SD \epsilon [5,\ 12.5]$), and three levels of skewness (negative skew, symmetric, positive skew). Ten ratings were presented in each trial on a VAS as described for the expectation task. The factorial combination of modality, stimulus intensity, cue mean, cue variance, and cue skewness resulted in a total of 48 combinations. Each combination was presented three times to each participant resulting in a total of 144 trials split into six blocks. Each block constituted a separate fMRI recording run and included 24 trials split into two mini-blocks of 12 trials each, in which only one modality was presented. The order of modality mini-blocks and the order of trials within modality were randomized for each participant. Skin conductance was recorded during the cued-perception task. Following the cued-perception task, participants completed a visual functional localizer task, in which they were presented with a visual checkerboard with varying contrasts. Data from this task were not used in any of the analyses reported in the current paper.

## Apparatus and recordings

**Heat stimulation.** An fMRI compatible Peltier thermode ($1.5 \times 1.5$ cm surface, PATHWAY CHEPS; Medoc, Inc, Israel) delivered heat to the left volar forearm of the participant in the stimulus-response task and the cued-perception task. The total stimulus duration was two seconds including ramp up and down from a 32°C baseline with extremely fast ramps (70°C/second up and 40°C/second down).

**Visual stimulation.** Radial checkerboards with a diameter of 12° visual angle flickered at a rate of 8 Hz on a 50% gray background. Screen luminance was calibrated using a Minolta LS-100 luminance meter. Stimulus presentation, thermode control, and response logging were implemented using the Psychophysics Toolbox (PTB-3, http://psychtoolbox.org/) [99].

## Behavioral data analysis

Analyses were performed with R version 4.3.1 (R studio version 2023.09.0 + 463). For reproducibility, we used the *checkpoint* package, which installs the R packages included in the code as they were on a specific date. We set the date to April 1, 2024. 211 out of 6480 trials in the cued-perception task were excluded due to technical issues with the pain device, rating device, or MRI scanner. Additional 234 trials were excluded because the ratings were too fast to represent deliberate ratings (response time < 0.2 second) or were not completed during the 4.5 seconds long rating period (response time > 4.5 seconds).

Linear mixed-effects models were fit to the perception (stimulus-response task and cued-perception task) and the expectation (expectation task) ratings with R's lmer function, with the packages "lme4" [100] and "lmerTest" [101]. All statistical tests were two-sided. In the stimulus-response task, we fitted two models, one for each modality, predicting the reported rating with the stimulus intensity level (numerical) as the regressor of interest, while controlling for the trial number (numerical). In the expectation task, included regressors were cue mean (numerical), cue variance level (binary), and cue skewness level (categorical), with all their interactions. We again fitted a separate model for each modality. In the cued-perception task, we included stimulus intensity level (binary), cue mean level (binary), cue variance level

(binary), and cue skewness level (categorical) as regressors (along with their interactions), and controlled for block and trial number (both numeric regressors). In all tasks, when significant interaction effects were found, we tested the related simple effects. All numeric variables (ratings, intensity level for the stimulus-response task, cue mean for the stimulus-response and expectation task, trial number for the stimulus-response and cued-perception tasks, and block number for the cued-perception task) were z-scored. Binary variables (modality, cue intensity level, cue variance level, and cue mean level in the cued-perception task) were modeled with one regressor, coded as 1 (high intensity level/ cue mean/ cue variance level) and -1 (low intensity level/ cue mean/ cue variance level). The cue skewness level was modeled with two regressors using the symmetric condition as the reference.

Moreover, in all models, participants were modeled as random effects. In each model, we started with a maximal random effects structure, modeling all random effects (intercept and slopes) and their correlations [102]. In case the model did not converge properly, we simplified the maximal model by first removing the random correlations and then reducing the random terms that indicated model converges issues (i.e., correlations of 1, or random variance of 0). Using these criteria preserves type I error while potentially increasing power when random effects estimates are near the boundary values [103].

Boundary effects on cue-based expectations were tested with mixed-effects models replicating Hird et al. [73]. The subjective stimulus value was computed as the averaged ratings of stimuli with the same intensity in the stimulus-response task, for each participant and modality. The cue PE (PE in [73]) was computed as the difference between the subjective stimulus value (sensory input) and the cue mean. Experienced PE (PE_sub in [73]) was computed as the difference between the reported rating and the subjective stimulus value. The "simple model" included experienced PE as the dependent variable (numeric), and cue PE, (cue PE)^2 and (cue PE)^3 (all numeric, non-centered) as predictors. The "complex" model included additional regressors and interactions: (cue PE)^4, trial number (per modality), and the interaction between each level of cue PE (cue PE, (cue PE)^2, (cue PE)^3, (cue PE)^4) and the trial number. Note that only a random intercept was included, since the model did not converge with random slopes. We report the results based on the complex model, but similar patterns were revealed with the basic model.

## Computational modeling

We developed a computational model to test how cue-based expectations are generated from the cue data, which consists of 10 values per cue. The model assumes that in each cue, each of these 10 values is weighted based on its relative location in the distribution of the 10 values (Fig 3; the model was inspired by Spitzer et al., 2017 [50]). The model yields a weight for each value, based on a combination of a power term modeling the weighting of inliers vs. outliers and a logistic term modeling the weighting of values that are smaller vs. larger than the mean. First, all 10 values of each cue were rescaled to [0,1] and then demeaned, such that cue values smaller than the mean were negative and cue values larger than the mean were positive. We denote the rescaled and demeaned value $X_i$ (where i ranges from 1 to 10 cue values).

Second, for each $X_i$, we computed a "power term" weight with the following equation:

$$Wk_i = \frac{sign(X_i) \ast |X_i|^k}{X_i} \tag{1}$$

Where $Wk_i$ is the power term weight of $X_i$, and $k \in (0, 1000]$ is a free parameter. When $k < 1$ inliers are over-weighted, when $k = 1$ outliers and inliers are equally weighted, and when $k > 1$ outliers are over-weighted. $Wk_i$ values for each cue where then normalized to [0,1] by dividing each $Wk_i$ by the sum of all $Wk$ weights for each cue $\sum_{i=1}^{10} Wk_i$.

Third, for each $X_i$, we computed a "logistic" weight using the following equation:

$$Wb_i = \frac{1}{1 + e^{(-b \ast X_i)}} \tag{2}$$

Where $Wb_i$ is the logistic weight of $X_i$, and $b \in$ [-1000, 1000] is a free parameter. When $b < 0$ values that are smaller than the mean are over-weighted, when $b = 0$ all values are equally weighted, and when $b > 0$ values that are larger than the mean are over-weighted. $Wb_i$ values are mathematically bounded between [0, 1].

Fourth, the two weights were combined, and normalized to the range [0,1], such that the weight of each $X_i$ was:

$$ W_i = \frac{Wk_i + Wb_i}{\sum_{i=1}^{10}(Wk_i + Wb_i)} \tag{3} $$

Finally, the expectation based on each cue was computed as the sum of each value $V_i$ multiplied by its weight $W_i$:

$$ Expectation = \sum_{i=1}^{10}(V_i * W_i) \tag{4} $$

Thus, the model computes an expectation value for each cue. The model was fitted to all trials (cues) of the expectation task of each participant. The free parameters $k$ and $b$ were optimized per participant and modality, based on ordinary least squares (OLS), with Matlab version 2022a *lsqcurvefit* function. We computed Pearson's correlation between observed (reported) expectations and model-based predicted expectations. Since the five different levels of cue mean contributed to the correlation and could artificially enhance the correlation between observed and predicted value, we also computed the correlation within each cue mean level. We then tested whether the optimized $k$ was significantly different from 1 and whether the optimized $b$ was significantly different from 0 at the group level, separately in each modality, with two-sided Wilcoxon signed rank tests. We also tested the correlation between the optimized $k$ for pain and for vision across participants, and the same for the optimized $b$, with a two-sided Spearman correlation test.

### Neuroimaging

**Neuroimaging data acquisition.** Data were collected on a 3 Tesla Siemens Trio MRI scanner with a 32 channels head coil at the University of Colorado Boulder Center for Innovation and Creativity. A high-resolution T1-weighted magnetization-prepared rapid gradient echo (MPRAGE) structural scan (0.8 × 0.8 × 0.8 mm voxels, TR: 2400 ms, TE 1: 2.07 ms, Flip angle: 8°, TI: 1200 ms, FoV Read: 256 mm) was performed on each participant at the beginning of the MRI session. We next acquired four images to compute B0-fieldmaps for distortion correction: two images with phase encoding in the anterior-posterior direction and two images with reversed phase encoding (2.7 × 2.7 × 2.7 mm voxels, TR: 7.22 s, TE: 73 ms, slices: 48, flip angle: 90°, FoV read: 220 mm). During the cued-perception task, a multiband echo-planar imaging (EPI) sequence (2.7 × 2.7 × 2.7 mm voxels, TR: 410 ms, TE: 27.2 ms, slices: 48, multiband factor = 8, flip angle: 44°, FoV read: 220 mm) was acquired. A single-band reference scan was acquired at the beginning of each run (block). We acquired 1250 volumes during each run of the cued-perception task and 776 volumes during the visual functional localizer task.

**Neuroimaging data preprocessing.** Structural and functional data were preprocessed using fMRIPrep version 20.2.3 (RRID:SCR_016216 [104,105]), which is based on Nipype 1.6.1 (RRID:SCR_002502 [106,107]).

**Anatomical data preprocessing.** The T1-weighted (T1w) image was corrected for intensity non-uniformity (INU) with N4BiasFieldCorrection [108], distributed with ANTs 2.3.3 (RRID:SCR_004757 [109]), and used as T1w-reference throughout the workflow. The T1w-reference was then skull-stripped with a *Nipype* implementation of the antsBrainExtraction.sh workflow (from ANTs), using OASIS30ANTs as target template. Brain tissue segmentation of cerebrospinal fluid (CSF), white-matter (WM) and gray-matter (GM) was performed on the brain-extracted T1w using fast (FSL 5.0.9, RRID:SCR_002823 [110]). Volume-based spatial normalization to one standard space (MNI152NLin2009cAsym) was performed through nonlinear registration with antsRegistration (ANTs 2.3.3), using brain-extracted versions of both T1w

reference and the T1w template. The following template was selected for spatial normalization: ICBM 152 Nonlinear Asymmetrical template version 2009c [[111], RRID:SCR_008796; TemplateFlow ID: MNI152NLin2009cAsym],

**Functional data preprocessing.** For each of the seven BOLD runs per subject (across all tasks and sessions), the following preprocessing was performed. First, a reference volume and its skull-stripped version were generated from the single-band reference (SBRef). Susceptibility distortion correction (SDC) was omitted. The BOLD reference was then co-registered to the T1w reference using flirt (FSL 5.0.9 [112]) with the boundary-based registration [113] cost-function. Co-registration was configured with nine degrees of freedom to account for distortions remaining in the BOLD reference. Head-motion parameters with respect to the BOLD reference (transformation matrices, and six corresponding rotation and translation parameters) are estimated before any spatiotemporal filtering using mcflirt (FSL 5.0.9 [114]). First, a reference volume and its skull-stripped version were generated using a custom methodology of fMRIPrep. The BOLD time-series were resampled onto their original, native space by applying the transforms to correct for head-motion. These resampled BOLD time-series will be referred to as preprocessed BOLD. The BOLD time-series were resampled into standard space, generating a preprocessed BOLD run in MNI152NLin2009cAsym space. Several confounding time-series were calculated based on the preprocessed BOLD: framewise displacement (FD), DVARS and three region-wise global signals. FD was computed using two formulations following Power (absolute sum of relative motions [115]) and Jenkinson (relative root mean square displacement between affines [114]). FD and DVARS are calculated for each functional run, both using their implementations in Nipype (following the definitions by [115]). All resamplings can be performed with *a single interpolation step* by composing all the pertinent transformations (i.e., head-motion transform matrices, susceptibility distortion correction when available, and co-registrations to anatomical and output spaces). Gridded (volumetric) resamplings were performed using antsApplyTransforms (ANTs), configured with Lanczos interpolation to minimize the smoothing effects of other kernels (Lanczos 1964). Non-gridded (surface) resamplings were performed using mri_vol2surf (FreeSurfer).

Many internal operations of fMRIPrep use Nilearn 0.6.2 ([116], RRID:SCR_001362), mostly within the functional processing workflow. For more details of the pipeline, see the section corresponding to workflows in fMRIPrep's documentation.

**Neuroimaging data analysis.** fMRI participant-level data processing was carried out using FEAT (FMRI Expert Analysis Tool) v. 6.00, part of FSL (FMRIB's Software Library, www.fmrib.ox.ac.uk/fsl) v. 6.0.4. Data were smoothed with a Gaussian kernel of 5 mm. A 100 hz high pass filter was used during first level analysis. The first level (run level) GLM model included two regressors per trial (48 regressors of interest per run in total): separate regressors for each cue period and for each stimulation period (single trial model, or a "Beta series" [117]). The global CSF signal and six motion parameters (translation and rotation each in three directions) were included as nuisance regressors. Variance Inflation Factor (VIF) was computed for each trial, and trials with VIF > 5 were excluded. This led to the exclusion of one trial (out of 6058 trials).

**Neuromarker and ROI analysis.** Group level analysis was performed with Matlab 2022a and 2023a, and CANlab tools (shared via Github at https://canlab.github.io/; also uses SPM12). A score for each neuromarker (NPS and SIIPS) was computed for each trial, once for the cue-evoked period and once for the stimulus-evoked period, based on the dot product between the trial's univariate map (from the single trial first-level analysis) and the neuromarker weight map. Similarly, we computed for each trial the mean activity across voxels of each of the individual a priori ROIs. For each brain measure (neuromarker or ROI), scores that were more than 3.5 SDs away from the mean score (of that same brain measure, period, and modality) were excluded. Overall, this resulted in exclusion of 0.4% of the scores (0.5% of the cue-evoked scores and 0.3% of the stimulus-evoked scores).

In order to allow comparison between the effects on the different brain measures, but keep the between-participants and within-participants effects unchanged, the scores of each brain measure were z-scored across all trials within the scope of the model (combination of modality [pain/ visual perception] and period [stimulus-evoked/ cue-evoked]) before including them as the dependent variable in the model (i.e., producing standardized beta estimates). Then, the z-scored

scores were tested with mixed-effects models, as described above for the expectation/ perceptual ratings (models were identical to the models described above, with the neuromarker/ ROI score replacing the outcome rating; see Methods section "Behavioral data analysis").

ROI analyses were conducted using a priori, planned comparisons in a limited number of predefined regions. Given that all tests were specified in advance and fully reported, multiple comparison correction was not strictly necessary [118,119]; however, to balance concerns related to Type I and Type II errors, we report both corrected and uncorrected results, by noting when a result is only significant with an uncorrected threshold (based on a Bonferroni correction within each set of regions). This approach ensures transparency while maintaining adequate statistical power to detect meaningful effects.

**Multilevel mediation analysis with neuromarkers.** We further tested, for each of the two neuromarkers, whether it mediated the effect of the cue-based expectation on the outcome rating, and whether it mediated the effect of the stimulus intensity on the outcome rating, each controlling for the other, for each modality separately. Brain mediation analysis tests whether a variable mediates the relationship between other variables, by identifying three statistical paths [9]. Path *a* captures the effect of the initial variable, usually the experimental manipulation (e.g., the cue-based expectation), on the mediator (e.g., the neuromarker score). Path *b* captures the effect of the mediator on the outcome variable (e.g., the outcome rating). Path *ab* captures the indirect effect of the initial variable on the outcome variable, i.e., the part of the relationship between the initial variable and the outcome variable that is formally mediated by the mediator. Path *c'* captures the direct effect of the initial variable on the outcome variable, that is not mediated by the mediator.

Here, we tested the effect of the cue-based expectations, or the stimulus intensity level, on the outcome rating, with the neuromarker (NPS/ SIIPS) as the mediator. Thus, four mediation models were tested for each modality: (1) Cue-based expectation (the trial-level expectation based on the presented cue and the expectation computational model with each participant's optimized $k$ and $b$ parameters) → trial-level NPS score → trial-level outcome rating, with trial-level stimulus intensity as a covariate; (2) Trial-level stimulus intensity → trial-level NPS score → trial-level outcome rating, with trial-level cue-based expectation as a covariate; (3) Same as model 1, but with SIIPS instead of NPS; (4) Same as model 2, but with SIIPS instead of NPS. In all four models, the neuromarker score, the cue-based expectation value, and the outcome rating were z-scored across all trials, separately for each modality. Each model was run with CANlab neuroimaging analysis tools' mediation.m function. Mediation was tested with bootstrapping, with 10000 samples [120].

## Supporting information

**S1 Text. Contains Methods, Results, Figs A-E and Tables A-K.**
(PDF)

## Acknowledgments

RB-N thanks the Golda Meir Fellowship and the Azrieli Foundation for their support.

## Author contributions

**Conceptualization:** Stephan Geuter, Tor D. Wager.

**Data curation:** Rotem Botvinik-Nezer, Stephan Geuter.

**Formal analysis:** Rotem Botvinik-Nezer.

**Funding acquisition:** Tor D. Wager.

**Investigation:** Stephan Geuter.

**Methodology:** Rotem Botvinik-Nezer, Stephan Geuter, Martin A. Lindquist, Tor D. Wager.

**Project administration:** Stephan Geuter.

**Software:** Rotem Botvinik-Nezer, Stephan Geuter.

**Supervision:** Tor D. Wager.

**Visualization:** Rotem Botvinik-Nezer.

**Writing – original draft:** Rotem Botvinik-Nezer.

**Writing – review & editing:** Stephan Geuter, Martin A. Lindquist, Tor D. Wager.

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
