## [Decision Letter · Decision Letter 0]

25 Nov 2024

PCOMPBIOL-D-24-01390Expectation generation and its effect on subsequent pain and visual perceptionPLOS Computational Biology Dear Dr. Wager, Thank you for submitting your manuscript to PLOS Computational Biology. After careful consideration, we feel that it has merit but does not fully meet PLOS Computational Biology's publication criteria as it currently stands. Therefore, we invite you to submit a revised version of the manuscript that addresses the points raised during the review process. Please submit your revised manuscript within 60 days Jan 25 2025 11:59PM. If you will need more time than this to complete your revisions, please reply to this message or contact the journal office at ploscompbiol@plos.org. Please include the following items when submitting your revised manuscript: * A rebuttal letter that responds to each point raised by the editor and reviewer(s). You should upload this letter as a separate file labeled 'Response to Reviewers'. This file does not need to include responses to formatting updates and technical items listed in the 'Journal Requirements' section below.* A marked-up copy of your manuscript that highlights changes made to the original version. You should upload this as a separate file labeled 'Revised Manuscript with Track Changes'.* An unmarked version of your revised paper without tracked changes. You should upload this as a separate file labeled 'Manuscript'. If you would like to make changes to your financial disclosure, competing interests statement, or data availability statement, please make these updates within the submission form at the time of resubmission. Guidelines for resubmitting your figure files are available below the reviewer comments at the end of this letter. We look forward to receiving your revised manuscript. Kind regards, Ming Bo CaiAcademic EditorPLOS Computational Biology Daniele MarinazzoSection EditorPLOS Computational Biology Feilim Mac GabhannEditor-in-ChiefPLOS Computational Biology Jason PapinEditor-in-ChiefPLOS Computational Biology  **Journal Requirements:**

At this stage, the following Authors/Authors require contributions: Rotem Botvinik-Nezer, Stephan Geuter, Martin A. Lindquist, and Tor D. Wager. Please ensure that the full contributions of each author are acknowledged in the "Add/Edit/Remove Authors" section of our submission form.

3) Please provide an Author Summary. This should appear in your manuscript between the Abstract (if applicable) and the Introduction, and should be 150u2013200 words long. The aim should be to make your findings accessible to a wide audience that includes both scientists and non-scientists. Sample summaries can be found on our website under Submission Guidelines:

5) We have noticed that you have uploaded Supporting Information files, but you have not included a list of legends. Please add a full list of legends for your Supporting Information files after the references list.

Potential Copyright Issues:

- Figures: 5A, 5B, and 6A. Please confirm whether you drew the images / clip-art within the figure panels by hand. If you did not draw the images, please provide a link to the source of the images or icons and their license / terms of use; or written permission from the copyright holder to publish the images or icons under our CC BY 4.0 license. Alternatively, you may replace the images with open source alternatives. See these open source resources you may use to replace images / clip-art:

7) Please amend your detailed Financial Disclosure statement. This is published with the article. It must therefore be completed in full sentences and contain the exact wording you wish to be published.

**Reviewers' comments:** Reviewer's Responses to Questions

**Comments to the Authors:**

Reviewer #1: In this manuscript, participants (N = 45) underwent a vicarious learning paradigm where they were presented with fictitious ratings prior to stimulus delivery to explore the effects of variability, skewness, and mean expectation on pain and visual perception. The authors found results generally aligned with contemporary Bayesian perception models, although some interesting deviations were found. These effects were mostly driven by the visual domain rather than the pain domain. The study presents an interesting approach, but it cannot be recommended for publication in its current form due to a number of major and minor issues (see below).

Major Issues:

1. The study design remains unclear in several aspects, such as the mention of “unreinforced trials” and the specifics of stimulus intensity levels and their relationship to the cue mean and participants’ pain ratings. As I understand it, their study fundamentally differs from Yoshida and our replication, where the mean of the vicarious pain ratings was directly related to the subjects' perception of the given stimulus. Here, however, it seems the authors used fixed cue mean values (30 and 70), ignorant of what the actual presented intensity was and how it should be perceived based on the individual psychometric curve. This could undermine the validity of the cues if they are too far removed from the participant’s actual perception, making them less credible. Additionally, details such as how many stimuli were presented and how often expectations were higher or lower only become partly clear at the end of the manuscript. I recommend that the authors revise the manuscript such that all relevant methodological information is presented before the results section.

2. The analysis of both modalities at once is confusing, especially given that the authors later on write that the pain and vision ratings are not directly comparable (making one questioning on the validity of these combined analysis then). Furthermore, while the interaction between modalities is often insignificant, the authors proceed to test them separately as if an interaction was present. This leads to considerable confusion, as many subsequent inferences about the implications for the pain field are based on general patterns across both modalities, even though these patterns do not hold when examining the pain data alone. I would propose focusing on the analysis separately per stimulus modality.

3. I struggled to interpret several of the results due to unclear descriptions of the models, all tested predictors, etc. In addition, in Fig. 4C, there are very strong Block (due to repositioning) and Trial effects (habituation) that remain unclear whether they are always accounted for in the analysis and whether results remain preserved if included. Furthermore, the computational models discussed at the end of several results sections add to the confusion, as they are only explained in the supplementary materials, and the conclusions drawn from individual-level model preferences or group-level model comparisons are not always in line with the conclusions in the context where the authors refer to them.

4. The authors presume that the only source of expectation is the fictitious ratings. While this is certainly an important source, as the experiment progresses, another significant source emerges—past experiences, which shape a different distribution of expected values. While the authors account somewhat for this by including a model where the weighting of the vicarious prior depends on prediction error (PE), they do not consider the possibility that expected pain could actually be a combination of recent pain experiences and social expectations. For instance, see (Jepma et al., 2018; Pavy et al., 2023). See also later points.

5. Variance, skewness, and cue mean were jointly manipulated, yet the authors tested them independently. The design does not ensure these variables are controlled for when testing one another, raising questions about the validity of the findings.

Minor Issues:

A) P6: Paragraph on effects of cue variance: The authors write “cue mean * modality interaction”—it is unclear whether the intention was to refer to cue variance instead.

B) P6: The sentence “Expectation ratings were significantly higher for lower cue variance in pain trials…but not in vision trials. In other words, participants expected more painful stimuli after more certain cues, but this was not true for noxious stimuli.” It is unclear if "noxious" was meant to be "visual. Else, I do not understand what the last part of this sentence implies?

C) P6: By the time the effects of skewness are described, the operationalization of skewness and the predictors tested remain unclear.

D) P7: The authors write: "In addition, expectation ratings were significantly higher in vision compared to pain trials. This result suggests that participants might have been biased by lower values particularly in the context of pain, or viewed themselves as less sensitive to pain compared to other participants.” It is unclear how this conclusion was reached, as an alternative explanation seems equally plausible—that participants were more susceptible to higher ratings in the vision condition. The basis for determining the direction of the effect is not clearly explained.

E) P10: When the authors mention correlating parameters with personality traits, it would be helpful to mention the final sample size to assess the meaningfulness of these analyses.

F) P10: The section on “The effects of cue-based expectation on perception” lacks sufficient information for the reader to fully understand this results section. Like how many intensity levels were presented and how the cue mean relates to participants’ pain ratings.

G) P11: The authors write:“ Participants rated stimuli with higher cue variance as more intense ( = 0.037, SE = 0.016 , t(5857) = 2.320, p = .020). This finding is consistent with Yoshida et al., (7), who suggested that uncertainty is aversive in the context of pain and thus lead to higher pain ratings, although a direct replication did not find this effect (30).” In the same paragraph is later stated that these effects only held for visual stimuli, undermining this initial conclusion.

H) P12: There are strong block effects (likely due to thermode repositioning) and within-block habituation effects in the pain data. It is unclear whether the authors controlled for these effects in their models, and if not, how these factors might have influenced the other findings. Additionally, in other parts of the analysis, the authors mention "trial effects," but it is not clear what exactly this refers to, requiring further clarification and why it was included in these analysis and not other analysis.

I) P13: The authors write: "Computational modeling strengthened the conclusion that most participants did not learn to ignore the cues during the task (see Supplementary Results - Computational Modeling of Perception), in line with previous studies (9,60–62)." It is unclear what exactly this conclusion is based on. Was this conclusion drawn from the model that incorporated the influence of prediction error (PE) on the weighting of cue expectations? As I understand it, this model was not preferred for the majority of individuals nor based group level model comparison, so it remains unclear how the authors arrived at this conclusion. Further explanation is needed.

J) P.21: The authors write: "Together with the assimilation towards predicted values in higher-level systems, our results largely support this working hypothesis, although we do not directly test encoding of different types of PEs (80)." It is unclear why the authors did not test this, as it could have been easily done. An expectation distribution combining both the vicarious information and participants' past pain ratings could have provided much more meaningful insights into how different types of expectations and information jointly shape pain experiences (see, for example, Jepma et al., 2018; Pavy et al., 2023)., where a similar dynamic approach was taken. This would enhance the understanding of the interplay between social and personal expectations in pain perception.

K) The authors strongly emphasize the Yoshida findings throughout the paper, but a recent meta-analysis (Pavy et al., 2024) has clearly demonstrated that uncertainty does not consistently lead to more pain. It would be advisable to portray this relationship in a more nuanced way, reflecting the broader evidence and avoiding overemphasis on a single set of findings.

L) P22-23: The authors write: "If participants over-weight extreme high-pain values—e.g., attend most to the most threatening potential values—that might explain why Yoshida et al. found increased pain with high-variance cues (7) while other studies have not found a main effect of uncertainty on pain (30,35). Here, we found that perceptual ratings were drawn towards larger extreme values, but this effect, like the effects of the cue variance, was significant mostly in visual perception and not in pain perception." There are two main issues with this:

i. It seems inconsistent to use this effect as an explanation for pain findings when it only held for visual stimuli in your own research. If the effect was not observed in the pain domain, it weakens the relevance of this explanation for the pain findings.

ii. The proposed explanation for Yoshida’s findings does not account for the broader literature. A well-powered replication study (Zaman et al., 2017) found no effect of uncertainty on pain perception despite using the identical paradigm, making the assumption that skewness drove their results unlikely. This needs to be addressed to provide a more balanced interpretation of the findings.

M) P23: The authors write: "Finally, the persistent effects of the nonreinforced cues in both pain and visual perception might represent modulation of early perceptual processes, modulation of perceptual decision-making." The repeated mention of "nonreinforced cues" is confusing (how I understood it they always received a stimulus during the cued-perception tasks) and suggests that I did not fully understood the task, even after reviewing the methods section multiple times. Further clarification is needed regarding the use and definition of these nonreinforced cues to ensure the task design is fully comprehensible.

N) P23: The authors write: "Beyond the assimilation to the mean value, reported expectations were also biased towards extreme values in both modalities, and specifically smaller values in pain." This effect seems to directly contradict the authors' presumed explanation of Yoshida’s findings. If skewness is driving the results, pain ratings should be lower in high variability conditions, which was not observed. This inconsistency needs to be addressed, as it undermines the author’s explanation linking extreme value biases with Yoshida’s findings.

O) P26: The authors write that 25 stimuli were presented in a randomized order, for a total of 50 stimuli. It is unclear from this sentence how many pain stimuli were actually presented—25 or 50? Later, it appears that 25 stimuli were used, based on 5 different intensities and 5 repetitions. This should be clarified

P) P27: Two levels of stimulation intensity are mentioned. Were these levels fixed or calibrated to the individual’s pain threshold, corresponding to specific pain ratings? This information is essential for understanding the methodology.

Q) P28: The authors write: “The rating cues were drawn from distributions with two means ( [30, 70]), two standard deviations ( [5, 12.5]), and three levels of skewness (negative skew, symmetric, positive skew).” Unlike Yoshida et al., where the vicarious ratings were directly related to the participants' expected pain ratings based on their psychometric function, it seems that here the ratings were unrelated. If this is correct, the authors should document how much the cued expectations differed between subsequent ratings, and whether large differences might have led to reduced credibility of the cued information.

R) P29: Please specify, for each predictor, whether it was categorical or continuous, and whether trial or block effects were included in the analysis.

S) P30: The description of the 5 different computational models is surprising, given that all the relevant analyses are presented in the supplementary information. Consider moving the model descriptions to the supplementary materials as well for consistency.

In conclusion, while the manuscript addresses an interesting topic, several aspects of the design, analysis, and interpretation need clarification to improve the clarity and robustness of the study. I enjoyed reading the manuscript and thinking about it. I hope the authors find my suggestions helpful.

Sincerely,

Jonas Zaman

References

Jepma, M., Koban, L., van Doorn, J., Jones, M., & Wager, T. D. (2018). Behavioural and neural evidence for self-reinforcing expectancy effects on pain. Nature Human Behaviour, 2(11), 838–855. https://doi.org/10.1038/s41562-018-0455-8

Pavy, F., Zaman, J., Van den Noortgate, W., Scarpa, A., von Leupoldt, A., & Torta, D. M. (2024). The effect of unpredictability on the perception of pain: a systematic review and meta-analysis. Pain, 165(8), 1702–1718. https://doi.org/10.1097/j.pain.0000000000003199

Pavy, F., Zaman, J., Von Leupoldt, A., & Torta, D. M. (2023). Expectations underlie the effects of unpredictable pain: a behavioral and electroencephalogram study. Pain. https://doi.org/10.1097/j.pain.0000000000003046

Zaman, J., Vanpaemel, W., Aelbrecht, C., Tuerlinckx, F., & Vlaeyen, J. W. S. (2017). Biased pain reports through vicarious information: A computational approach to investigate the role of uncertainty. Cognition, 169, 54–60. https://doi.org/10.1016/j.cognition.2017.07.009

Reviewer #2: This paper addresses multiple different conflicts in literature and thoroughly discusses them through experiments, models and neural analysis:

- They define expectation experiment to understand how people’s expectations are shaped by varying levels of mean, variance, and skewness

- This experiment, specially the skewness experiment allowed them to investigate the influence of presence and direction of outliers

- They further investigated the influence of the above on pain and visual perception

- Additionally exploring the pain and visual domain allowed them to identify similarities and differences of influence of expectation on perception in different modalities

- Finally they investigated how neural responses during pain and visual perception are affected by the different properties of the cue and by the cue-based expectations

The paper does a great job of detailing the models and the experiment and elaborately connect results to the literature in the discussion section

There are some questions that I need some clarification on and have some small comments for improvements:

- Introduction first paragraph says: “However, which brain signals assimilate to expectations, perhaps forming a neural substrate for perception, and which encode PEs and thus drive learning, is a topic of active investigation.” I would like authors to add some appropriate citations

- Introduction says “may stem from differences across experimental designs ….” It would be good to summarize the follow-up statements into a table and also additionally mentioning how their work resolves/addresses some of them in a new column. Since this paper has multiple points to make, a summary discussion is easy to get the take away message.

- Introduction says “Furthermore, this kind of nonlinear weighting might depend on the sensory modality …” : It would be good to provide some reference and better motivate the study of pain modality. Additionally I would like to better understand a summary of what the literature says for other modalities and how pain and vision compare to that. I also did not find a clear motivation to using these two modalities

- I am curious to understand I am trying to understand how the expectation generation is different from updating the parameters of prior distribution in Bayesian inference, particularly is this any different from tuning/learning hyperparameters in Bayesian statistics?

- The authors have a computational model for weighting of cue values, but from the methods I am sure I understand how they made sure the weights were not biased by the outliers? logistic regression is supposed to be influenced by outliers, so how did the authors ensure the overweighting of outliers was not an artifact? ref: results section “We found that participants significantly over-weighted outliers in both modalities..”

- Figure 3A, I am confused by the end points of the plot for k=100. In Figure 3B, I am worried about Simpson’s paradox, can the authors comment on that a little. Figure 3C, this is across all subjects, correct? Are there any errors bars? Is the result significant? Also in general there is a lack of figure reference in text in conclusive paragraphs in results section. The statement “participants overweight extreme cue values in both modalities, and demonstrate an optimism bias specifically in pain, relying more on low than high extreme values.” is that based on Figure 3 c alone? Like mentioned before, additional figure references in such conclusions will help readers.

- The authors put too much parameter details in the text which is hard to follow when reading, it would be great to have references in form of tables. The authors already have multiple tables in the supplementary and they is easy to read and follow.. Example in Results section: “However, all correlations between the available measures in our sample (Fear of Pain score (56), Pain Catastrophizing score (57), and State-Trait Anxiety Inventory score (58)) and optimized k or b for each modality across participants, were not significant ….” has too much parameters added which is hard to follow and conclude the findings.

- In Figure 4C I am not sure I understand why the plot looks zagged. Does the error bar represent across subject variability?

- I would also like to understand within subject variability of the expectations and perceptions, specifically if it is more for vision versus pain?

- Minor: Is there a typo in this sentence? “This result suggests that participants might have been biased by lower values particularly in the context of pain, or viewed themselves as less sensitive to pain compared to other participants “

Reviewer #3: Summary and general evaluation:

The manuscript at hand explores the significant topic of how expectations influence perception in pain research through a computational approach. While it is clear that expectations have an influence on the perception of pain, the effect of its associated uncertainty is still not investigated in detail and led to conflicting evidence in the research literature. One issue that might be at the heart of it is the influence of outliers and skewness of cues that could lead to a biased way of building up expectations. The used paradigm presents distributions of cues as pain intensity and visual contrast ratings by previous participants and proves to be extremely useful to investigate such effects. The authors use both conventional linear mixed models and an innovative computational model to incorporate these influences on expectations to explain their expectation rating data. They reveal some interesting aspects that have not been explored previously, for example that participants over-weight extreme values in the cue distribution when they construct their expectations. In detail, they found a domain-independent over-weighting of outliers and a pain-specific over-weighting of values smaller than the cue mean value like an optimism-bias. This is a real strength of the manuscript, as it provides valuable starting points for further investigations into context, paradigms and populations in which such results would persist or differ, e.g., clinical populations with chronic pain.

In addition, the authors explore the subsequent effect on the perception process. In a separate experiment session, the participants received similar visual cues followed by real stimuli and had to rate the intensity of their percept. They also use a linear mixed model to fit the responses and find that it is mainly the used intensity and the cue mean that influence these subsequent responses. Only for visual stimuli they find some additional effects of cue variance and skewness on perception. The authors then check for stimulus intensity and cue effects in the brain by testing the concurrently recorded fMRI data for such trial-wise patterns. Whereas their previously developed neuromarker for pain (NPS) mainly activated in direction of stimulus intensity, their neuromarker for intensity independent processed captured the influence of the cue mean, which nicely fits expectations about the nature of these biomarkers. They add analyses for different ROIs to describe the effect of the cues on different perceptual processing stages. Additionally, they present a computational perceptual model that builds upon the expectation model but only refer to its results in the supplement. Overall, the figures clearly convey their messages and the manuscript is overall well written. The computational models are explained in detail and simulation figures enhance the understanding of their functionality. The authors provide additional data and code in their github repository that seems well maintained and accessible. The manuscript is generally properly placed in the context of the literature. The manuscript is of high value and clear importance especially to the pain community as predictive processing theories need to be formally tested using appropriate models before they find their way into clinical applications. I have a few issues I would like to see addressed before I could recommend the manuscript for publication (sorted from more major to more minor points).

1. The most severe point that hindered my understanding and evaluation of some of the results was the lack of description of the used intensities in the perceptual task. Whereas in the stimulus intensity task five different temperatures/contrasts are used, the authors describe the use of two intensities in the perception task but as far as I could tell, they do not describe how these were chosen. Depending on this, the evaluation of results in this task could drastically change. Predictive processing theories that predict an integration of cue information into the perception only hold true if the discrepancy between cue and stimulus information is not too strong. A highly discrepant stimuli will have a weaker influence on the perception as previously demonstrated by Hird et al., 2019, Nature Scientific Reports.

It would also help in evaluating how “threatening” the stimuli really were, as this could potentially influence how large cue values are interpreted. Knowing how far apart the two different intensities are would also help to evaluate how “ambiguous” the pain stimuli really were, as the authors mention several times that they are more ambiguous than the visual stimuli, yet do not provide information on that. This would also change how informative the cues then were to the participants. If the used intensities are that far apart that it becomes clear to many participants that they are actually just deciding between two competing hypotheses they might abandon the cues altogether, especially as they are not informative about the upcoming stimulus as the authors describe. It would be nice to not only see these points addressed but also to maybe see results about that in the figure 4 about the perception task. It would be great to see for example separate lines for the different intensity levels in 4C to see both the effects of stimulus intensity and cue mean as these effects are the strongest in these analyses.

2. Please check your paragraph about effects of cue variance on expectations on page 6. You describe a significant cue mean x modality interaction. Do you mean cue variance x modality instead or a three-way-interaction? If the former, then one of the following sentences should also be saying that participants expected more intense stimuli after more certain cues, but that this was not true for visual stimuli. Additionally, the line of argumentation on why the cue mean and variance interaction should show up in the stimulus ratings but not in the expectations are a bit unclear, on page 11 you describe this much better.

3. Please provide information on statistical thresholds used for the ROI analyses. Which significance level is used and how is the correction for multiple comparisons done?

4. In figure 2A, why were the five cue mean levels collapsed and which ones were collapsed? Could you also provide the full figure on this?

5. Please also provide information on the blinding of the participant towards the paradigm structure. I assume they were not aware of the fact that there are only two or five intensities in each task involving stimuli? And did they believe that the presented cue distributions were real ratings? Because depending on this belief, they might integrate this information more or less strongly.

6. Which thermode did you use for the heat stimulation? You mention the PATHWAY ATS but then describe very fast ramp-up times of 70°C/s, which to my knowledge are associated with the PATHWAY CHEPS model if I am not mistaken. Or do you have a newer model that can provide such timings?

7. Please refrain from comparing nociceptive and visual stimuli for example on page 13 when you discuss your supplementary results. As you correctly describe in the limitations section the scales of the stimuli are not comparable, and a stronger assimilation towards cued values does not necessarily mean that pain perception in general is more ambiguous but that the chosen pain stimuli used in this study were probably more ambiguous than the visual stimuli. When matching the ambiguity of the stimuli one could maybe see the same effects. The same can be said in response to your discussion of modality-specific vs modality-general effects on higher-level processing ROIs on page 23. Maybe the use of more ambiguous visual cues would lead to similar activations in higher-level processing regions as for the nociceptive stimuli.

8. Please clearly provide separate statements when you describe your expectation and your perception results, as it should be immediately clear to the reader that these experiments were done on a different day. For example, both in the abstract and in the discussion, you switch between both descriptions back and forth, so that one could be tempted to draw conclusions from one to the other. Yet I feel like there are different processes at work, and the discrepancy is interesting in itself. It would be interesting to hear also about your opinions regarding the discrepancy of expectation and perception effects. Are the effects of variance or skewness just to small to then have a notable effect when the expectation is weighted against the stimulus input? Or do participants learn over time to ignore the cues? Or do they use more elaborate models in a context where they are asked to rate the expectation but if the task demand is different, they use more simple heuristics to evaluate the percept? You tried to tackle some of these questions also in your supplementary analyses. I was expecting to see these in the main manuscript as the methods descriptions for that are also in the main manuscript, maybe move one of these. But then I also understand why they cannot capture all the questions one might have regarding the two processes of expectation and perception. One major issue for me in these models is the inability to capture the effect of cue variance on the integration of cue information as you also mention this already in the abstract as one of the main goals of the paper. This integration should be trial-specific yet the weightings of input and expectation are modelled the same for all conditions, or only decaying over time. It is also not so clear to me what can be learned in a paradigm in which the cues are non-informative. Nonetheless, it was great to see how different individuals’ data fitted the models so differently which raises even more interesting questions.

9. Could you provide data on the scaling factors described on page 32? It would be interesting to see how the different day and context changes the ratings especially for pain as I would expect some stronger fluctuations within an individual than in vision.

10. Please make it clearer that the brain imaging results are based on the regressor for the stimulation. This only becomes evident after reading that you also modelled the cue period in the supplement data.

11. Did the participants experience the cues already with real stimuli when they were practicing the cued-perception task outside the scanner? Can you describe how long they practiced?

12. The last sentence in the abstract seems a bit strong/general to me, as you mention the effects of expectations would operate on higher-level processes, yet you mentioned in your discussion that this could also completely differ when using a different type of expectations. Especially, when using non-informative, non-reinforced cues I would expect cues to have less of an impact on lower-level processes, but would not completely dismiss the idea of possible effects there. You also mention in this sentence that selective attention would provide a better fit to your data, yet this is not mentioned in your manuscript. Can you elaborate on that?

13. I was a bit puzzled about the results of the computational model of expectations in figure 3B. To me it seems that the model for visual stimuli predicts somewhat distinct clusters of responses at the position of the cue means. Why is that? Is it something in the participant’s rating behavior that is different for the visual expectation task, maybe because they use some knowledge of the previous stimulus intensity task? I would expect comparable results for both modalities on naïve participants, yet their previous experience of the stimuli and scale use has changed their behavior in the expectation task. Even the model performance is better in pain data. Why is that?

14. Why did you not detect a prediction error response in the anterior insula as in previous investigations (e.g., Geuter et al., 2018, eLife; Fazeli & Büchel 2018; Journal of Neuroscience)? Is it mainly because the cue vs stimuli relation is much different in these paradigms and participants in these studies do not integrate the cue information into their percept but are mainly showing a strong mismatch between the cue information and the stimulus input?

15. In figure 4C you nicely describe that the first trial of each run shows an increased average pain rating because a new skin site is used. As you also exclude two training trials in the previous stimulation task, it might be reasonable to also exclude the first trials here? Have you tried this? I feel like it as a huge influence on the results and might increase the noise in your data when it is not being modelled.

16. Just out of curiosity and because we sometimes acquire similar data: do you have the skin conductance results available? Are they also shifted towards the cue mean or only showing effect of stimulus intensity?

**Have the authors made all data and (if applicable) computational code underlying the findings in their manuscript fully available?**

Reviewer #1: Yes

Reviewer #2: Yes

Reviewer #3: Yes

PLOS authors have the option to publish the peer review history of their article (what does this mean? ). If published, this will include your full peer review and any attached files.

**Do you want your identity to be public for this peer review?** For information about this choice, including consent withdrawal, please see our Privacy Policy .

Reviewer #1: **Yes: ** Jonas Zaman

Reviewer #2: No

Reviewer #3: No

 **Figure resubmission:**While revising your submission, please upload your figure files to the Preflight Analysis and Conversion Engine (PACE) digital diagnostic tool, https://pacev2.apexcovantage.com/. PACE helps ensure that figures meet PLOS requirements. To use PACE, you must first register as a user. Registration is free. Then, login and navigate to the UPLOAD tab, where you will find detailed instructions on how to use the tool. If you encounter any issues or have any questions when using PACE, please email PLOS at figures@plos.org. Please note that Supporting Information files do not need this step. If there are other versions of figure files still present in your submission file inventory at resubmission, please replace them with the PACE-processed versions. 
---

## [Decision Letter · Decision Letter 1]

14 Apr 2025

Dear Dr. Wager,

We are pleased to inform you that your manuscript 'Expectation generation and its effect on subsequent pain and visual perception' has been provisionally accepted for publication in PLOS Computational Biology.

Best regards,

Ming Bo Cai

Academic Editor

PLOS Computational Biology

Daniele Marinazzo

Section Editor

PLOS Computational Biology

The editor thinks that Reviewer 1's critiques were also well addressed.

Reviewer's Responses to Questions

**Comments to the Authors:**

Reviewer #2: The authors did a thorough job responding to my comments. Here's a summary of how they addressed the feedback:

1. Overall Clarity and Contribution: Reviewer 2 appreciated the paper's scope, rigor, and relevance. The authors strengthened the paper’s clarity by:

* Adding missing citations where needed (e.g., on brain signals encoding prediction errors).

* Including a new summary Table 1 in the Introduction that contrasts previous studies with their own, helping readers grasp the contributions at a glance.

* Better motivating the use of pain and visual modalities, explaining how each relates to predictive coding theories and prior literature.

* Clarifying the definition of expectation generation versus Bayesian prior updates and improving terminology (now using “cue-based expectations”).

2. Methodological Clarifications:

* They clarified their computational modeling choices, emphasizing that outlier weighting was not due to statistical artifacts and that logistic regression was not the method used for that part.

* They improved the figure legends and interpretations, including more figure references in the results to support conclusions about, for instance, optimism bias in pain.

3. Terminology and Conceptual Framing:

* They replaced ambiguous terms like “expectation generation” with “cue-based expectation generation” throughout the manuscript to better reflect what was modeled.

* They acknowledged conceptual overlap between Bayesian updating and expectation generation but distinguished their approach from classic Bayesian parameter learning.

4. Data Visualization and Interpretation:

* They addressed concerns about Simpson’s paradox in their Figure 3B and confirmed that correlations remain significant within each cue mean level.

* They clarified how outliers in cue distributions were treated in modeling and showed these were designed-in, not statistical flukes.

Future Work: The authors clarified the difference of how cue-based expectations differ from full Bayesian learning with hyper-parameter tuning. While this work didn’t explore hybrid models that might combine cue-based and experience-based learning, it would be a interesting future work.

The authors responded thoughtfully and thoroughly. They improved clarity, grounded their methods more firmly in the literature, and showed they took the reviewer’s feedback seriously. The paper is now much stronger.

Reviewer #3: All my suggestions have been adequately and thoroughly addressed and I have no concerns with the manuscript in its current form.

**Have the authors made all data and (if applicable) computational code underlying the findings in their manuscript fully available?**

Reviewer #2: Yes

Reviewer #3: Yes

PLOS authors have the option to publish the peer review history of their article (what does this mean? ). If published, this will include your full peer review and any attached files.

**Do you want your identity to be public for this peer review?** For information about this choice, including consent withdrawal, please see our Privacy Policy .

Reviewer #2: No

Reviewer #3: **Yes: ** Ulrike Horn

---

## [Editor Report · Acceptance letter]

PCOMPBIOL-D-24-01390R1

Expectation generation and its effect on subsequent pain and visual perception

Dear Dr Wager,

I am pleased to inform you that your manuscript has been formally accepted for publication in PLOS Computational Biology. Your manuscript is now with our production department and you will be notified of the publication date in due course.

With kind regards,

Anita Estes
